# RoboMoRe: LLM-based Robot Co-design via Joint Optimization of Morphology and Reward

## Abstract

Robot co-design, the joint optimization of morphology and control policy, remains a longstanding challenge in the robotics community. Existing approaches often converge to suboptimal designs because they rely on fixed reward functions, which fail to capture the diverse motion modes suited to different morphologies. We propose **RoboMoRe**, a large language model (LLM)-driven framework that integrates morphology and reward shaping for co-optimization within the robot design loop. RoboMoRe adopts a dual-stage strategy: in the coarse stage, an LLM-based *Diversity Reflection* mechanism is proposed to generate diverse and high-quality morphology–reward pairs and *Morphology Screening* is performed to reduce unpotential candidates and efficiently explore the design space; in the fine stage, top candidates are iteratively refined through alternating LLM-guided updates to both reward and morphology. This process enables RoboMoRe to discover efficient morphologies and their corresponding motion behaviors through joint optimization. The result across eight representative tasks demonstrate that without any task-specific prompting or predefined reward and morphology templates, RoboMoRe significantly outperform human-engineered design results and competing methods. Additional experiments demonstrate robustness of RoboMoRe on manipulation and free-form design tasks.

## 1 Introduction

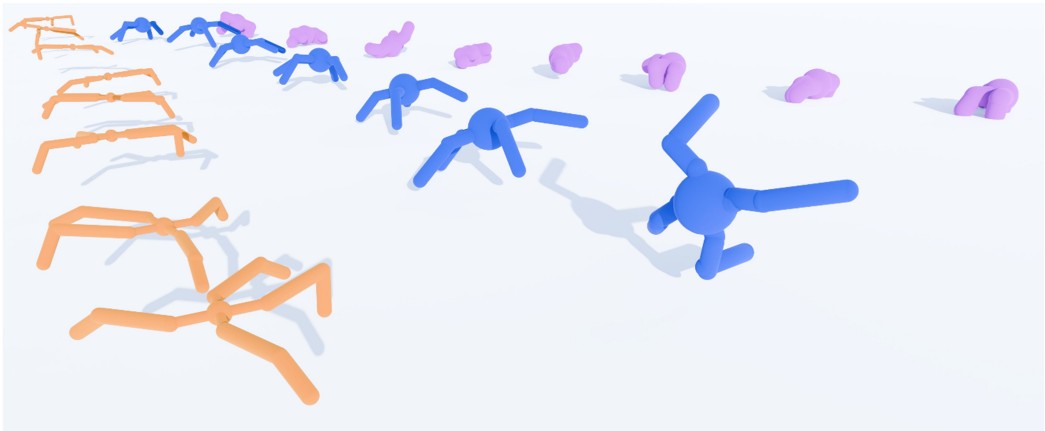

Figure 1: **Diverse motion behaviors and robot morphologies generated by RoboMoRe.** Orange: A bio-inspired long-legged robot crawling forward in a climber posture. Blue: A uniform robot advancing by jumping. Purple: A low-center-of-gravity robot rolling forward along the ground.

One of the main goals of artificial intelligence is to develop effective approaches for the creation of embodied intelligent systems Steels & Brooks (2018). Inspired by natural organisms, where body structure and brain are two key factors for completing any task in a real environment, a successful intelligent robot typically requires concurrently optimizing its structure design and control mechanism

Saridis (1983). Such a co-design problem has been a long-standing key challenge in the robotics and machine learning communities Wang et al. (2023); Ma et al. (2021); Xu et al. (2021).

Although many existing approaches have achieved promising results, they typically rely on a fixed reward function, which significantly limits the potential of robot co-design Lu et al. (2025); Zhao et al. (2020). Under such constraints, robots are only capable of learning a narrow range of motion behaviors tied to a single, static objective. For robots with varying morphologies, a single, uniform reward function severely constrains improvements in locomotion performance and the exploration of diverse motion modalitiesCui et al. (2025); Wang (2024); Han et al. (2024).

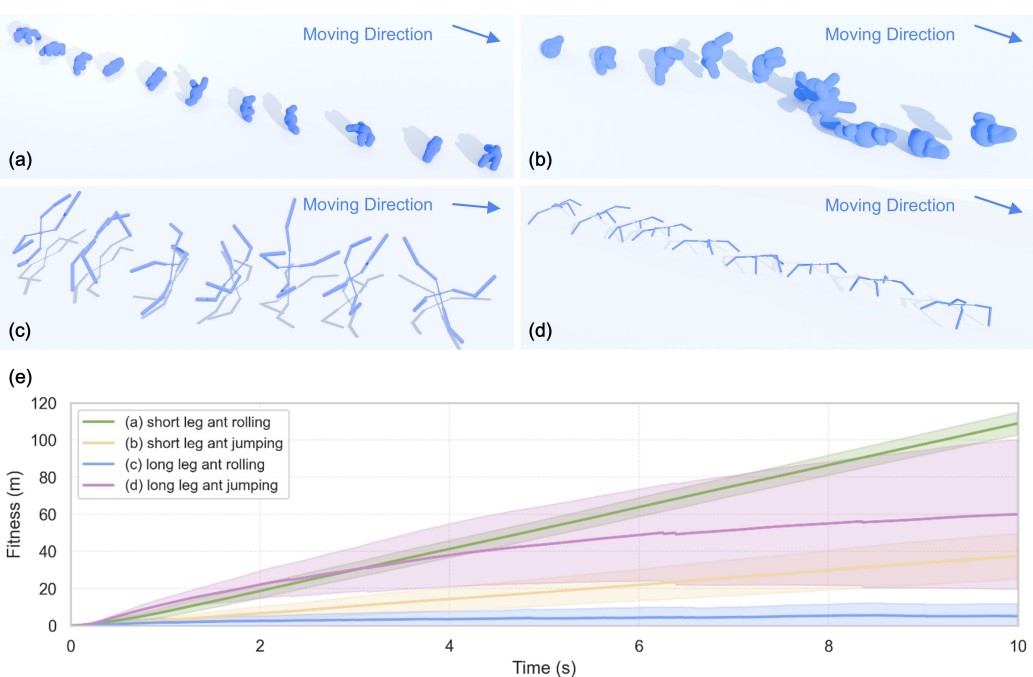

Figure 2: **Examples of short-leg and long-leg ants exhibiting rolling and jumping behaviors.** (a–d) Representative motion behaviors across the four design cases. (e) Time-series fitness (walking distance) over 100 independent evaluations for each design case.

To address this limitation, we introduce **RoboMoRe**—a novel LLM-based robot co-design framework that jointly optimizes both **Robo**t **Mo**rphology and **Re**ward functions. The core philosophy of RoboMoRe is to identify optimal reward functions tailored to each robot morphology. By tailoring reward functions to match specific morphologies, we can unlock a wider variety of motion behaviors and boost overall performance. For example, as shown in Figure 2, a long-legged robot might benefit from a reward that encourages jumping, while a low-profile robot might perform better with a reward that promotes rolling (Appendix D).

Yet, optimizing morphology and reward simultaneously is far from trivial: coupling the two dramatically enlarges the design space, making it difficult to guarantee both optimization quality and speed. To address this challenge, RoboMoRe adopts a Coarse-to-Fine optimization paradigm. In the coarse phase, we introduce a general *Diversity Reflection* mechanism to enhance exploration of the joint space and mitigates the risk of premature convergence to local optima. In parallel, we observe a morphology-dominant bottleneck property: poor morphologies impose a hard cap on efficiency that reward shaping cannot overcome (see Appendix G.3.1). Leveraging this insight, RoboMoRe performs a *Morphology Screening* mechanism to effectively screening out unpromising morphologies and accelerating subsequent optimization. In the fine phase, the most promising candidates are iteratively refined using an alternating optimization scheme Bezdek & Hathaway (2003): the LLM first adjusts the reward to better suit the current morphology, then updates the morphology accordingly. This alternating cycle ensures steady convergence toward high-performing morphology–reward pairs, while maintaining computational efficiency.

In summary, our contributions are:

1. We propose RoboMoRe, a general co-design framework that leverages LLM to jointly optimize robot morphology and reward functions.

2. We adopt a Coarse-to-Fine optimization paradigm, where the coarse stage leverages Diversity Reflection and Morphology Screening to promote broad exploration of the design space while efficiently pruning low-potential candidates, and the fine stage employs alternating refinement to precisely optimize the remaining morphology–reward pairs.

3. Experiments across eight representative design tasks show that RoboMoRe uncovers novel morphologies and motion modalities, significantly outperforming both human-designed and competing automated methods, with a 2.1× speedup and a 1.8× reduction in cost. Additional experiments further demonstrate its robustness on manipulation and free-form design tasks.

## 2 RELATED WORK

**Robot Co-design**  Early evolutionary robotics demonstrated that co-evolving morphology and control policies outperform sequential design, highlighting the interdependence of "body" and "brain" in agents Bravo-Palacios et al. (2020). Early work framed robot co-design as a graph search problem and addressed using the advanced evolutionary algorithms, yielding progressively more intriguing result Sims (2023); Ha (2019); Pathak et al. (2019); Wang et al. (2019); Zhao et al. (2020). However, as both kinematic complexity and task difficulty grow, approaches such as evolutionary algorithms, Bayesian optimization, and deep reinforcement learning must contend with vast design spaces that demand substantial computational resources and more efficient search strategies, thereby constraining their development. Recently, large language models (LLMs) hold significant promise for transforming the robot design process thanks to their strong in-context learning capabilities and extensive prior knowledge. However, research on LLM-based robot co-design remains limited to only a few early exploratory studies Zhang (2024); Qiu et al. (2024); Lehman et al. (2023); Song et al.. Among these, LASER Song et al. has shown promising results in designing soft voxel robots (SVR), particularly with regard to reflecting diversity. However, it still relies on carefully crafted task-specific prompts and a highly narrow method for diversity reflection which can only be applied to SVR, greatly limits its generalizability to other types of robots. In contrast, our work introduces highly general prompts and a more flexible approach to diversity reflection. Most importantly, existing methods share a critical limitation with fixed reward strategies, which significantly hinders their effectiveness in robot co-design.

In general, while these methods demonstrate potential, previous approaches have not incorporated reward shaping. By fixing the reward function and only optimizing robot morphology—i.e., *Fixed Reward, Varying Morphology*—they have restricted the possibilities of true co-design. In our paper, we argue that reward shaping can encourage different robots to develop distinct and diverse movement strategies, ultimately resulting in superior performance.

**Reward Shaping**  Reward shaping is a pivotal component in the development of robotic control strategies. Early reward engineering methods predominantly relied on manual trial-and-error tuning Knox et al. (2023) and inverse reinforcement learning (IRL) Ho & Ermon (2016). Manual tuning is time-consuming and requires substantial domain expertise, whereas IRL demands costly expert demonstrations and frequently produces opaque, black-box reward functions, thereby limiting its practical applicability. Subsequent researches have investigated automated reward optimization via evolutionary algorithms Faust et al. (2019); Chiang et al. (2019), but these efforts were typically limited to task-specific implementations that only tuned parameters within predefined reward templates. Related diversity-oriented RL methods focus on discovering varied behaviors under a fixed embodiment, but they do not address structural diversity and therefore have limited applicability to morphology-level design in co-design settings Eysenbach et al. (2018). More recently, large language models (LLMs) have been used to generate reward functions for new tasks. For instance, Eureka Ma et al. (2023) provides a general framework that, without any task-specific prompting or pre-defined reward templates, can generate reward functions that outperform human-engineered ones. Similarly, Text2Reward Yang (2024) shows that LLM-based reward shaping can produce symbolic, interpretable rewards and enable robots to learn novel locomotion behaviors (*e.g., Hopper backflip, Ant lying down*). Beyond reward-centric methods, Co-Imitation Rajani et al. (2023) demonstrates that

morphology and control can be jointly optimized through imitation learning; however, its objective is based on expert demonstration matching rather than on generating or shaping task rewards.

Inspired by these advances, we introduce LLM-driven reward shaping into robot co-design. By generating diverse reward functions with LLMs, RoboMoRe enables robots with varying morphologies to discover novel motion strategies, significantly enhancing their performance.

Table 1: **Comparison of RoboMoRe with related literature.**

| Literature | Method | Mor. Design | Reward Shaping | Diversity | Iterations | Optimum |
|---|---|---|---|---|---|---|
| EvoGym Bhatia et al. (2021) | BO, EA | ✓ | ✗ | N/A | Slow | Local |
| RoboGrammar Zhao et al. (2020) | Graph Search | ✓ | ✗ | N/A | Slow | Local |
| RoboMorph Qiu et al. (2024) | LLM | ✓ | ✗ | Off | Fast | Local |
| LASER Song et al. | LLM | ✓ | ✗ | On | Fast | Local |
| Eureka Ma et al. (2023) | LLM | ✗ | ✓ | Off | Fast | Local |
| **RoboMoRe (Ours)** | LLM | ✓ | ✓ | On | Fast | Near Global |

## 3 PROBLEM DEFINITION (ROBOT CO-DESIGN)

The Robot Co-design problem involves jointly optimizing a robot's morphology and control policy to maximize its performance in a given environment. It is defined as a tuple $P = \langle \mathcal{M}, \Theta, \mathcal{R}, \mathcal{A}, F \rangle$, where $\mathcal{M}$ is the environment model, $\Theta$ is the robot morphology design space, and $\mathcal{R}$ is the space of reward functions. $\mathcal{A}(\theta, R) : \Theta \times \mathcal{R} \to \Pi$ is the co-design algorithm that takes a robot design $\theta$ and reward function $R$, then learns an optimal policy $\pi$. $F$ is the fitness function which measures real-world performance but can only be accessed through policy execution. Existing works aim to solve this problem via searching the optimal robot design $\theta^*$ that lead to the highest performance given a fixed reward function $R_0$:

$$\theta^* = \arg \max_{\theta \in \Theta} F(\pi_{\theta, R_0}), \quad (1)$$

However, this can lead to local optimum due to limitation of $R_0$. To address this, our objective is to jointly optimize the robot morphology $\theta^*$ and reward function $R^*$ that lead to the global optimal performance:

$$\theta^*, R^* = \arg \max_{\theta \in \Theta, R \in \mathcal{R}} F(\pi_{\theta, R}), \quad (2)$$

where the optimized policy is obtained by $\pi_{\theta, R} := \mathcal{A}(\theta, R)$.

## 4 METHOD

In this section, we formally introduce the RoboMoRe pipeline (Fig. 3), which follows a Coarse-to-Fine optimization paradigm. The coarse stage leverages Diversity Reflection and Morphology Screening to generate diverse morphology–reward candidates and prune unpromising ones, ensuring broad yet efficient exploration of the design space (Sec 4.1). The fine stage then iteratively refines the most promising candidates through an alternating optimization scheme, where the LLM adjusts reward functions and morphologies in turn until convergence on high-performing design pairs (Sec 4.2).

### 4.1 COARSE OPTIMIZATION WITH DIVERSITY REFLECTION AND MORPHOLOGY SCREENING

The goal of coarse optimization is to approximate the distribution of morphology-reward pairs near the global optimum. Traditional approaches typically rely on random sample (*e.g.*, sampling initial candidates from gaussian distribution) Ha (2019); Bhatia et al. (2021), which although ensures diversity, often fails to generate samples sufficiently close to the global optimum. As a result, these methods require hundreds of thousands of optimization steps to converge, making them computationally expensive and inefficient. To address this, RoboMoRe leverage LLM with Diversity Reflection to generate high-quality and diverse samples, coupled with a Morphology Screening mechanism ensuring effciency.

RoboMoRe leverages LLMs to generate initial samples. For morphology design, the task description and a masked robot structure file (*e.g.*, MJCF XML) are provided to the LLM, which then generates a novel set of design parameters. All morphology-related parameters (*e.g.*, limb lengths) are masked to eliminate human priors, ensuring fair and unbiased comparisons. For reward design, RoboMoRe

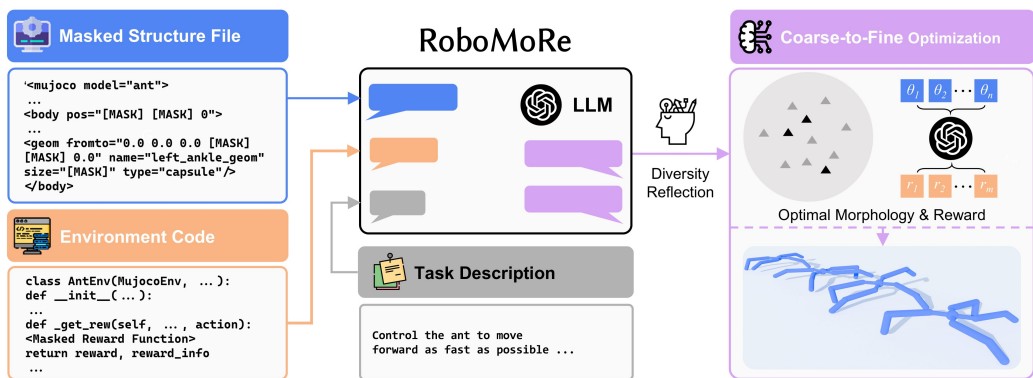

Figure 3: **The overall pipeline of RoboMoRe**. The LLM receives the task description, environment code, and a masked robot structure file as input. The environment code specifies simulation variables for reward design, while the masked structure file defines incomplete morphology parameters to be completed by the LLM. Our algorithm adopts a coarse-to-fine strategy: in the coarse phase, the LLM leverages Diversity Reflection to generate a broad set of morphology–reward samples and Morphology Screening to filter unpotential candidates; in the fine phase, morphologies and rewards are alternately refined, converging toward an optimal morphology–reward pair.

ingests the raw environment source code—excluding any predefined reward functions—as context, enabling the LLM to synthesize task-specific reward functions that are executable in Python and directly compatible with the simulator. A strict output format enforces structural consistency and correctness. To ensure scalability across diverse tasks, RoboMoRe employs a highly generalizable prompting strategy (Appendix A).

Although LLMs can produce high-quality samples by leveraging their strong reasoning and generation capabilities, without explicit mechanisms to promote diversity they often produce repetitive outputs and converge to suboptimal local optima (Table 4). This challenge is non-trivial, as simply increasing the sampling temperature leads to extremely slower generation, syntactically invalid outputs, and still fails to prevent repetitive motion behaviors (Appendix I). To address this, we propose *Diversity Reflection*, a general and effective mechanism in which the LLM reflects on previously generated samples and deliberately produces new candidates that maximize diversity relative to past designs. *Diversity Reflection* is task-agnostic and can be seamlessly extended to other domains without modification. Experimental results in Sec. 6.3 demonstrate that it consistently improves both sample diversity and optimization performance across diverse tasks.

In parallel, the enlarged design space introduced by joint optimization increases computational overhead and can hinder optimization speed. Hence, RoboMoRe introduces *Morphology Screening* to enhance optimization efficiency. This mechanism is grounded in the observation of morphology-dominant bottleneck property: suboptimal morphologies impose an inherent ceiling on efficiency that cannot be alleviated by reward shaping (Appendix G.3.1). Specifically, we observe that the first LLM-generated reward function generally lacks motion-specific priors, making it a suitable prior for assessing morphology quality. It is therefore used to evaluate all morphology candidates with the reinforcement learning algorithm, after which only the top-performing morphologies are retained for subsequent iterations, while low-performing ones are excluded from further reward evaluations. By discarding such morphology candidates early, RoboMoRe reduces redundant computations and accelerates optimization without compromising final performance.

## 4.2 Fine Optimization with Morphology Reward Alternating

In fine optimization, we adopt an alternating optimization strategy that iteratively refines both components toward the global optimum. Top candidates from the coarse stage are provided to the LLM, which derives *textual gradients* ($\frac{\partial_{LLM} f}{\partial \theta}$, $\frac{\partial_{LLM} f}{\partial r}$) as inferred optimization directions, and uses them to progressively improve morphology and reward along this momentum. Meanwhile, several promising coarse-stage results are retained to broaden the search space and uncover additional optimization opportunities. Please see Appendix H for the algorithm details.

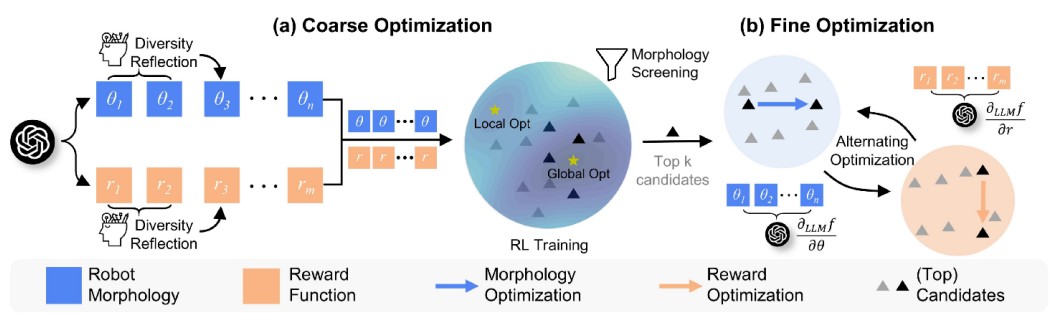

Figure 4: **Overview of Coarse-to-Fine algorithm.** (a) In the coarse optimization stage, the LLM first generates diverse and high-quality reward functions and robot morphologies using a Diversity Reflection mechanism. These reward–morphology pairs are trained via reinforcement learning, after which a Morphology Screening step filters out unpromising designs to ensure efficiency. (b) In the fine optimization stage, the LLM employs alternating optimization to jointly refine both the robot's reward and morphology, leading to an optimized design and control strategy.

## 5 EXPERIMENT SETUP

**Environments**  We implemented our environments using the MuJoCo Gym simulator Towers et al. (2024), covering eight distinct robot design tasks: Ant, Ant-Powered, Ant-Desert, Ant-Jump, Hopper, Half-Cheetah, Swimmer and Walker. For morphology design, we mask certain parameters in the MuJoCo XML files to prevent the LLM from gaining prior knowledge of the robot's design. For reward function design, the LLM is provided with the official environment code with the reward functions removed. Regarding task descriptions, we use the official descriptions from the environment repository whenever available. Detailed information on robot design morphologies, reward functions, and task descriptions can be found in Appendix E.

**Baselines**  We consider 4 strong baselines to highlight performance of RoboMoRe: 1) **Bayesian Optimization (BO)** is a classic algorithm designed to optimize expensive-to evaluate functions Snoek et al. (2012); Rasmussen (2003). It employs a probabilistic model (*e.g.* Gaussian Process) as a surrogate for the objective function and determines where to sample based on predicted mean and uncertainty. 2) **Eureka** is a general LLM-based reward shaping framework Ma et al. (2023). Similar to an evolutionary algorithm, It uses an LLM to generate a large number of samples, select the best-performing ones, and acts as a heuristic operator for mutation. 3) **Eureka (Mor.)** is a variant of the Eureka framework in which the original reward shaping prompt is replaced with RoboMoRe's morphology design prompt (Appendix A.1.2), enabling the LLM to design robot morphology. 4) **Human** represents the template morphology and reward functions from official Mujoco website Towers et al. (2024).

**Evaluation Metrics**  In all experiments, we use a normalized metric **efficiency** to evaluate performance, which is defined as division of fitness and robot volume. This choice is crucial for stimulating the LLM's material-aware design capabilities rather than brute-force scaling. We compute volume using custom scripts for each type of robot. While fitness is also reported (*e.g.*, distance for locomotion, height for jumping tasks), **efficiency** serves as the core metric in our optimization iterations. Please refer to Appendix F for more implementation details.

## 6 RESULTS

### 6.1 COMPARISON WITH COMPETING METHODS

**RoboMoRe outperforms human design robots and competing methods.**  Table 2 shows that RoboMoRe consistently generates highly performant and efficient designs, significantly surpassing both human-designed and baseline methods (Fig. 5). The performance gap is most striking in efficiency, as humans rarely approach near-optimal designs for complex tasks. Competing methods

underperform mainly because Eureka and Eureka (Mor.) neglect diversity, producing repetitive suboptimal offspring, and both focus narrowly on either reward or morphology. Section 6.3 and Section 6.2 analyze these aspects. Robustness tests on Ant-Powered (gear power variation) and Ant-Desert (terrain variation) further confirm RoboMoRe's advantages, with details in Appendix G.4 and G.5.

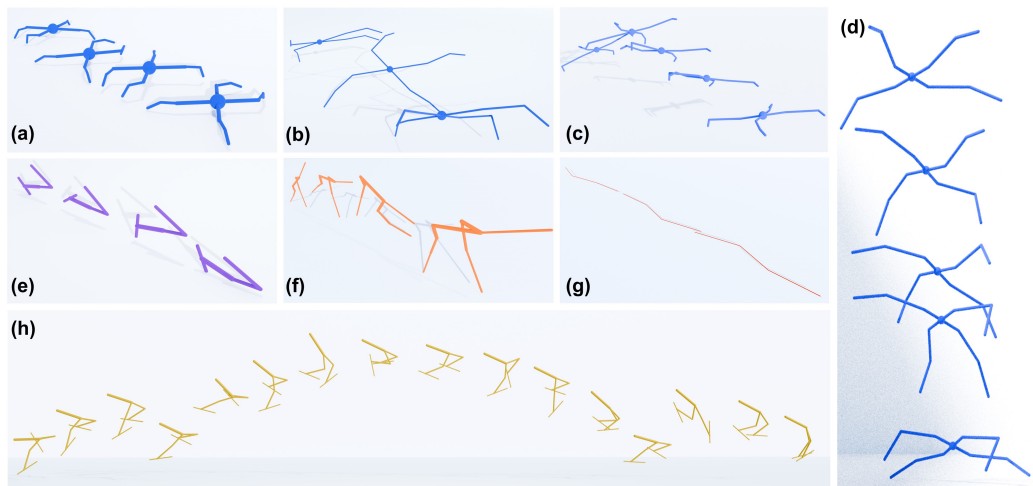

Figure 5: **Optimal designs generated by RoboMoRe.** (a) Ant. (b) Ant-Powered, where motor power is doubled. (c) Ant-Desert, with the terrain replaced by desert. (d) Ant-Jump, where the task is changed from locomotion to jumping. (e) Hopper. (f) Half-Cheetah. (g) Swimmer. (h) Walker.

Table 2: **Performance comparison of human and competing methods across eight tasks.** We report both efficiency and fitness metrics. **Note that we set efficiency as the optimization goal instead of fitness in implementation.**

| Metric | Method | Ant | Ant-Powered | Ant-Desert | Ant-Jump | Hopper | Half-Cheetah | Swimmer | Walker |
|---|---|---|---|---|---|---|---|---|---|
| **Efficiency ↑** **(Fitness/Volume)** | Bayesian Optimization | 707.13 | 74.52 | 223.61 | 3.53 | 128.00 | 4242.89 | 57.53 | 111.95 |
| | Eureka | 121.54 | 24.91 | 49.82 | 7.60 | 428.11 | 12,157.92 | 23.96 | 139.73 |
| | Eureka (Mor.) | 5,203.82 | 1,160.94 | 2.51 | 229.55 | 609.04 | 15,531.26 | 16,335.19 | 1,946.60 |
| | Human | 68.22 | 26.13 | 29.84 | 6.83 | 433.69 | 11,975.34 | 18.86 | 170.29 |
| | **RoboMoRe** | **31,038.41** | **32657.10** | **36,995.77** | **902.76** | **3,776.66** | **495,373.71** | **57,627.40** | **6,665.85** |
| **Fitness↑** | Bayesian Optimization | **390.14** | 41.11 | **100.49** | 1.95 | 2.75 | 194.82 | 8.66 | 2.54 |
| | Eureka | 8.81 | 4.54 | 9.08 | 1.38 | 6.77 | **257.55** | 2.56 | 3.92 |
| | Eureka (Mor.) | 143.07 | 41.52 | 0.08 | 5.33 | 1.61 | 145.09 | 89.04 | 2.91 |
| | Human | 12.43 | 4.76 | 5.44 | 1.24 | 6.86 | 253.69 | 2.01 | **4.78** |
| | **RoboMoRe** | 165.18 | **70.30** | 39.60 | **1.48** | **15.10** | 135.71 | **109.35** | 4.17 |

## 6.2 EFFECTIVENESS OF REWARD SHAPING AND MORPHOLOGY DESIGN

**Reward Shaping can improve performance by inspiring morphology-suited motion behaviors.** Table 3 shows that reward shaping improves performance across most tasks by aligning rewards with morphology-specific locomotion strategies. The effect is particularly strong in tasks with rich morphological parameterization: for example, Half-Cheetah achieves nearly a twofold improvement in efficiency when reward shaping is applied. By contrast, simpler tasks such as Swimmer, with only a few parameters and a low-dimensional action space, exhibit limited benefits. These findings indicate that reward shaping provides the greatest gains when morphology is highly parameterized and the state space more complex.

Table 3: **Ablation study of RoboDesign across efficiency metrics under different Reward (Rew.) and Morphology (Mor.) configurations.**

| Method | Metric | Ant | Ant-Powered | Ant-Desert | Ant-Jump | Hopper | Half-Cheetah | Swimmer | Walker |
|---|---|---|---|---|---|---|---|---|---|
| **Efficiency ↑** | RoboDesign (w/o Mor. & Rew.) | 21.64 | 8.68 | 20.69 | 6.08 | 94.19 | 4,984.63 | 6.82 | 15.36 |
| | RoboDesign (w/o Mor.) | 97.74 | 18.04 | 15.79 | 279.42 | 561.61 | 7,945.81 | 15.63 | 22.37 |
| | RoboDesign (w/o Rew.) | 6,190.39 | 3,026.53 | 3,742.68 | 6.34 | 1,653.59 | 67,992.31 | **22,009.94** | 1401.21 |
| | **RoboDesign (Full)** | **10,464.17** | **6,537.45** | **8,001.67** | **396.45** | **1,951.44** | **129,440.98** | 21,931.15 | **1,482.60** |

**Morphology Design still plays a fundamental role.** Although reward shaping can enhance performance by encouraging novel motion patterns, morphology design remains crucial. Removing morphology optimization (w/o Mor.) leads to drastic efficiency drops. In Half-Cheetah, efficiency decreases by more than one order of magnitude compared to the full model, and similar or even larger degradations appear in tasks such as Walker and Ant-Powered. These results indicate that poor morphologies impose hard bottlenecks that reward shaping alone cannot overcome. Variation in material usage across designs further explains the disparities in efficiency, highlighting that template morphologies are far from optimal and that morphology co-optimization is indispensable.

## 6.3 ANALYSIS OF DIVERSITY AND COMPUTATIONAL EFFICIENCY

Table 4: **Performance comparison of RoboMoRe variants in terms of efficiency, morphology diversity, and reward diversity.** For clarity, only results from the coarse optimization stage are reported.

| Metric | Method | Ant | Ant-Powered | Ant-Desert | Ant-Jump | Hopper | Half-Cheetah | Swimmer | Walker |
|---|---|---|---|---|---|---|---|---|---|
| Efficiency ↑ | RoboMoRe (Random Sample) | 12.91 | 11.83 | 12.91 | 3.70 | 86.39 | 1,451.44 | 116.44 | 42.17 |
| | RoboMoRe (w/o Diversity Reflection) | 136.91 | 45.56 | 52.44 | 32.36 | 111.91 | 1,536.94 | 2,338.88 | 87.54 |
| | **RoboMoRe (w/ Diversity Reflection)** | **487.13** | **398.60** | **151.52** | **49.41** | **375.52** | **9,557.07** | **4,607.74** | **149.54** |
| Morphology Diversity (Coefficient of Variation) ↑ | RoboMoRe (Random Sample) | 0.51 | 0.50 | 0.49 | 0.48 | **0.55** | 0.99 | 0.41 | **0.60** |
| | RoboMoRe (w/o Diversity Reflection) | 0.44 | 0.46 | 0.57 | 0.44 | 0.53 | 0.82 | 0.29 | 0.43 |
| | **RoboMoRe (w/ Diversity Reflection)** | **0.64** | **0.70** | **0.64** | **0.66** | 0.54 | **1.38** | **0.61** | 0.50 |
| Reward Diversity (Self-BLEU) ↓ | RoboMoRe (w/o Diversity Reflection) | 0.70 | 0.67 | 0.70 | 0.57 | 0.70 | 0.75 | 0.74 | 0.72 |
| | **RoboMoRe (w/ Diversity Reflection)** | **0.50** | **0.46** | **0.46** | **0.48** | **0.47** | **0.45** | **0.43** | **0.42** |

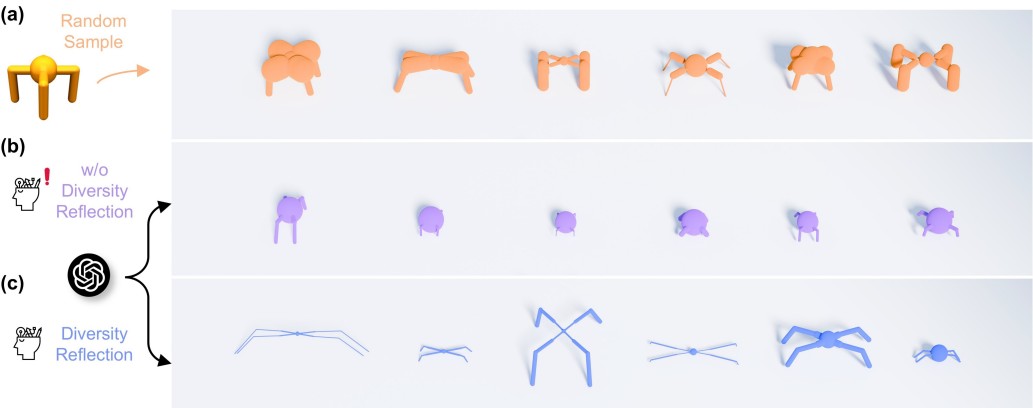

Figure 6: **Qualitative results among (a) Random Sample, (b) RoboMoRe (w/o Diversity Reflection), (c) RoboMoRe.** It can be observed that RoboMoRe with Diversity Reflection could generate more high-quality and diverse robot morphology examples compared to others.

**RoboMoRe produces highs quality and diverse morphology parameters and reward functions.** We compare our method against two baselines: (1) Random Sample (RS), a commonly used strategy in Evolutionary Algorithms (EA) and Bayesian optimization (BO), where initial designs are sampled from a Gaussian distribution Frazier (2018); Bäck & Schwefel (1993), and (2) a setting without Diversity Reflection (w/o DR). Morphology diversity is quantified using the **coefficient of variation** Bedeian & Mossholder (2000), while reward diversity is assessed via the **Self-BLEU** metric Zhu et al. (2018). All diversity metrics are averaged over fifty generated samples. Table 4 illustrates that the Diversity Reflection mechanism substantially improves both the performance of generated samples and the diversity of rewards and morphologies.

While Random Sample introduces diversity, it often generates low-quality designs—such as bulky and unbalanced legs for the Ant robot, see Fig. 6. This low initial quality explains why traditional methods such as EA and BO typically require thousands of iterations to converge. In contrast, LLMs, even without diversity reflection, are capable of generating higher-quality samples thanks to their inherent reasoning and design capabilities. However, without explicitly promoting diversity, such methods fail to explore more performant regions of the design space, resulting in suboptimal performance. RoboMoRe achieves the best overall performance because it effectively leverages Diversity Reflection to generate samples that are both high-quality and diverse, enabling a more efficient and robust design

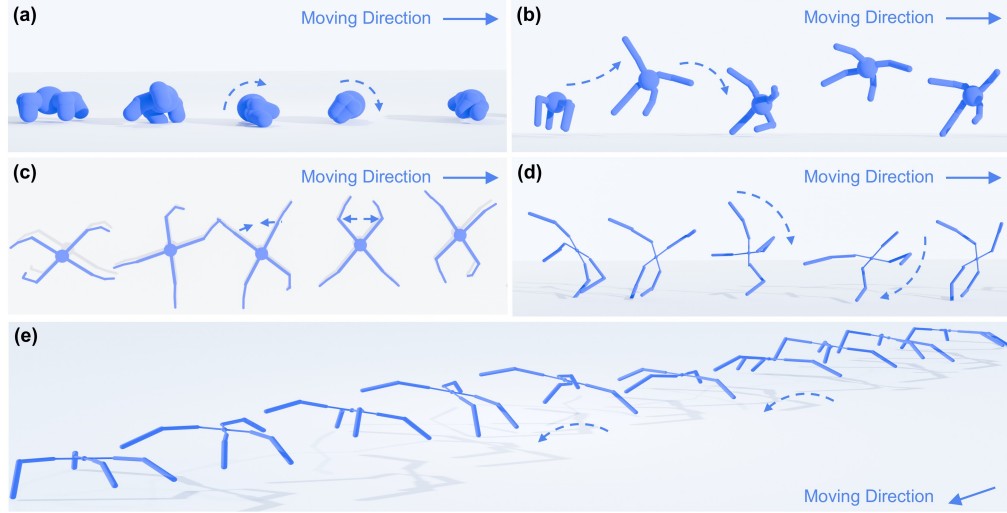

Figure 7: **Qualitative results of diverse and novel motion behaviors uncovered by RoboMoRe.** (a) The short-legged, low-center-of-gravity robot rolls forward along the ground. (b) The uniform robot propels itself forward with sideways hops. (c) The crab-inspired robot repositions itself and scuttles forward in a crab-walking manner. (d) The agile, lightweight robot advances by executing a side flip. (e) The agile, long-legged robot moves forward with a light hopping gait.

optimization process (Fig. 7). Additional comparison on Diversity Reflection with LaSER can be found in Appendix G.3.

**Computational efficiency of RoboMoRe.** Table 5 compares both computational runtime and LLM token cost across methods. Each iteration takes about 15 minutes on an RTX 4080 Super GPU with a 24-core CPU, so Eureka requires about 20 hours (80 iterations) and BO about 25 hours (100 iterations) to complete one design task. In contrast, RoboMoRe finishes in roughly 10 hours (41 iterations), cutting runtime by about 50%. In terms of LLM token usage, Eureka consumes about 170–200K tokens ($2.0)

Table 5: **Computational efficiency comparison.**

| Method | Tokens (Cost) | Avg Iter. |
|---|---|---|
| Eureka (Mor.) | 199.3K ($2.11) | 80 |
| Eureka | 169.3K ($2.03) | 80 |
| BO | N/A | 100 |
| RoboMoRe | 105.6K ($1.13) | 41 |

per run, while RoboMoRe requires only 105K tokens ($1.1), also reducing token expenditure by 50%. This efficiency stems from RoboMoRe's modularized generation and Morphology Screening. First, unlike Eureka which issues $16 \times 5 = 80$ queries per run, RoboMoRe requires only 25 morphology and 5 reward queries. Second, Morphology Screening further accelerates optimization by filtering unpromising samples early. Please see details on Appendix G.3.1.

## 6.4 ADDITIONAL RESULTS ON MANIPULATION AND FREE-FORM DESIGN

We further evaluate RoboMoRe on two additional and challenging benchmarks: (i) high-DOF manipulation (DM Control Tunyasuvunakool et al. (2020)), where a robot arm with twenty-three tunable morphology parameters is tested across four manipulation tasks (Insert peg, Insert ball, Bring peg, Bring ball), and (ii) free-form soft-voxel robotics design (EvoGym Bhatia et al. (2021)) across four tasks (Walker-v0, Carrier-v0, Pusher-v0, Jumper-v0). Results from Table 6 reveal that reward shaping and morphology design contribute complementary benefits: removing either component leads to substantial performance drops, highlighting the necessity of morphology–reward co-optimization for discovering efficient and diverse robotic behaviors. Further experiments on free-form soft-voxel robotics demonstrate robustness of RoboMoRe. Details are provided in Appendix G.1 (manipulation) and Appendix G.2 (free-form soft-voxel robotics).

Table 6: **Performance comparison on DM-Control manipulator tasks.** Results are averaged over 100 runs.

| Method | Insert peg | | | Insert ball | | | Bring peg | | | Bring ball | | |
|---|---|---|---|---|---|---|---|---|---|---|---|---|
| | Fitness ↑ | Efficiency ↑ | Success % ↑ | Fitness ↑ | Efficiency ↑ | Success % ↑ | Fitness ↑ | Efficiency ↑ | Success % ↑ | Fitness ↑ | Efficiency ↑ | Success % ↑ |
| RoboMoRe (w/o rew.) | 10.20 | 29,154.94 | 15 | 19.28 | 80,857.70 | 34 | 20.54 | 66,966.59 | 29 | 18.35 | 36,258.80 | 34 |
| RoboMoRe (w/o mor.) | 10.22 | 6,702.98 | 17 | 16.47 | 10,792.47 | 24 | 14.38 | 9,424.37 | 22 | 16.47 | 11,238.10 | 24 |
| Human | 6.35 | 4,165.91 | 10 | 15.66 | 10,265.19 | 24 | 10.41 | 6,824.57 | 12 | 8.90 | 11,238.10 | 24 |
| **RoboMoRe** | **14.20** | **43,958.53** | **18** | **24.26** | **89,646.18** | **59** | **24.97** | **96,157.95** | **34** | **24.26** | **56,261.08** | **50** |

## 6.5 ABLATION OF COARSE-TO-FINE OPTIMIZATION.

**Both coarse and fine stages contribute significantly to co-design performance.** Fig. 8 illustrates the complementary roles of the coarse and fine stages in achieving optimal co-design. The coarse stage explores the joint morphology–reward space broadly, producing diverse candidates, while the fine stage refines them through alternating updates. In high-dimensional or complex tasks, coarse exploration alone cannot deliver near-optimal designs, making fine refinement essential. Conversely, in simpler environments such as Swimmer and Hopper, the fine stage alone can match or slightly surpass the coarse stage. Overall, the full two-stage pipeline consistently achieves the best performance across tasks.

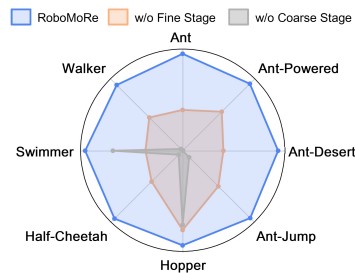

Figure 8: **Comparison of efficiency for RoboMoRe, w/o Fine Stage, and w/o Coarse Stage.**

## 7 CONCLUSION

In conclusion, we present RoboMoRe, an LLM-driven framework that integrates reward shaping into the robot co-design process. By adopting a coarse-to-fine optimization strategy—combining Diversity Reflection and Morphology Screening with alternating LLM-guided refinement—RoboMoRe successfully discovers novel, diverse and high-performing morphology-reward pairs. Experiments demonstrate our method consistently outperforms human baselines and prior co-design approaches. We hope RoboMoRe contributes to a deeper understanding of morphology and reward joint optimization in robot design and serves as a springboard for future innovations at the intersection of generative AI and embodied intelligence.

**Ethics statement** This work does not involve human or animal subjects and therefore does not raise concerns related to IRB approval or human data privacy. All experiments were conducted on publicly available simulation environments, and we provide full details of task settings and preprocessing in the appendix to ensure transparency. No proprietary or sensitive data were used, and no conflicts of interest or external sponsorships influenced this research. We have carefully reviewed the ICLR Code of Ethics and affirm that our work complies with its principles of fairness, transparency, research integrity, and responsible dissemination.

**Reproducibility statement** We have taken several steps to ensure the reproducibility of our results. All algorithmic details of RoboMoRe, including the Coarse-to-Fine optimization pipeline, Diversity Reflection, Morphology Screening mechanisms, hyperparameters, training configurations, and evaluation metrics are provided in Appendix E and F. To facilitate replication, we include detailed prompts and pseudocode for LLM interactions in Appendix A and H. The additional environments used (DM Control and EvoGym) are publicly available and discussed in details, and we provide full details of task settings and preprocessing steps in Appendix G.1 and G.2.

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

APPENDIX OUTLINE

# A FULL PROMPTS

## A.1 COARSE OPTIMIZATION PROMPTS

We take the Ant as an example to illustrate the prompt used in coarse optimization. The prompt for morphology design mainly consists of three parts: the task description, a masked MuJoCo XML file, and the required output format. The MuJoCo XML is masked to prevent LLMs from seeing the human-designed structure in advance. For brevity, we show only part of the MuJoCo XML file and environment code. Please refer to MuJoCo Gym for more details Towers et al. (2024).

### A.1.1 PROMPTS FOR REWARD SHAPING

```
You are a reward engineer trying to write reward functions to solve reinforcement learning
tasks as effectively as possible.
Your goal is to write a reward function for the enviroment that will help the agent learn the
task described in text.

Task Description: The ant is a 3D quadruped robot consisting of a torso (free rotational
body) with four legs attached to it, where each leg has two body parts.
The goal is to coordinate the four legs to move in the forward (right) direction by applying
torque to the eight hinges connecting the two body parts of each leg and the torso (nine body
parts and eight hinges), You should write a reward function to make the robot move as faster
as possible.

Here is the environment codes:
class AntEnv(MujocoEnv, utils.EzPickle):
<Enviroment Code>

A template reward can be:
    def _get_rew(self, x_velocity: float, action):
        <reward function code you should write>
        return reward, reward_info

The output of the reward function should consist of two items:
(1) 'reward', which is the total reward.
(2) 'reward_info', a dictionary of each individual reward component.

The code output should be formatted as a Python code string: '```python ...```'.

Some helpful tips for writing the reward function code:
(1) You may find it helpful to normalize the reward to a fixed range by applying
transformations like 'numpy.exp' to the overall reward or its components.
(2) Make sure the type of each input variable is correctly specified and the function name is
"def _get_rew():"
(3) Most importantly, the reward code's input variables must contain only attributes of the
provided environment class definition (namely, variables that have the prefix 'self.'). Under
no circumstances can you introduce new input variables.
```

### A.1.2 PROMPTS FOR MORPHOLOGY DESIGN

```
Role: You are a robot designer trying to design robot parameters to increase the fitness
function as effective as possible.
Your goal is to design parameters of robot that will help agent achieve the fitness function
as high as possible.
Fintess function: walk distance/material cost.
Task Description: The ant is a 3D quadruped robot consisting of a torso (free rotational
body) with four legs attached to it, where each leg has two body parts. The goal is to
coordinate the four legs to move in the forward (right) direction by applying torque to the
eight hinges connecting the two body parts of each leg and the torso (nine body parts and
eight hinges).
Here is the xml file:
"""
<mujoco model="ant">
...
  <worldbody>
    <light cutoff="100" diffuse="1 1 1" dir="-0 0 -1.3" directional="true" exponent="1"
pos="0 0 1.3" specular=".1 .1 .1"/>
    <geom conaffinity="1" condim="3" material="MatPlane" name="floor" pos="0 0 0" rgba="0.8
0.9 0.8 1" size="40 40 40" type="plane"/>
    <body name="torso" pos="0 0 {height}">
      <camera name="track" mode="trackcom" pos="0 -3 0.3" xyaxes="1 0 0 0 0 1"/>
      <geom name="torso_geom" pos="0 0 0" size="{param1}" type="sphere"/>
      <joint armature="0" damping="0" limited="false" margin="0.01" name="root" pos="0 0 0"
type="free"/>
```

```
        <body name="front_left_leg" pos="0 0 0">
          <geom fromto="0.0 0.0 0.0 {param2} {param3} 0.0" name="aux_1_geom" size="{param8}"
type="capsule"/>
          <body name="aux_1" pos="{param2} {param3} 0">
            <joint axis="0 0 1" name="hip_1" pos="0.0 0.0 0.0" range="-40 40" type="hinge"/>
            <geom fromto="0.0 0.0 0.0 {param4} {param5} 0.0" name="left_leg_geom"
size="{param9}" type="capsule" />
            <body pos="{param4} {param5} 0" >
              <joint axis="-1 1 0" name="ankle_1" pos="0.0 0.0 0.0" range="30 100"
type="hinge"/>
              <geom fromto="0.0 0.0 0.0 {param6} {param7} 0.0" name="left_ankle_geom"
size="{param10}" type="capsule"/>
            </body>
          </body>
        </body>
...
"""
</mujoco>
Attention:
1. For reducing material cost to ensure the effieciency of robot design, you should reduce
redundant paramters and increase paramters who control the robot.
2. Your design should fit the control gear and others parts of robots well.
Output Format:
Please output in json format without any notes:
{ "parameters": [<param1>, <param2>, ..., <param10>],
"description": "<your simple design style description>", }
#   param1 is the size of the torso, positive.
#   param2 is is the end attachment x point of leg, positive.
#   param3 is is the end attachment y point of leg, positive.
#   param4 is is the end attachment x point of hip, positive.
#   param5 is the end attachment y point of hip, positive.
#   param6 is the end attachment x point of ankle, positive.
#   param7 is the end attachment y point of ankle, positive.
#   param8 is the size of the leg, positive.
#   param9 is the size of the hip, positive.
#   param10 is the size of the ankle, positive.
```

### A.1.3   PROMPTS FOR DIVERSITY REFLECTION

```
Please propose a new morphology design, which can promote high fitness function and is quite
different from all previous morphology designs in the design style. (Morphology)
```

```
Please write a new reward function to encourage more robot motion behaviours, which can
promote high fitness function and is quite different from all previous reward functions in
the design style. (Reward)
```

## A.2   FINE OPTIMIZATION

### A.2.1   PROMPTS FOR ALTERNATING REFINEMENT

```
There are also some design parameters and their fitness. Please carefully observe these
parameters and their fitness, try to design a new parameter to further improve the fitness.
(Morphology)
```

```
There are also some reward functions and their fitness. Please carefully observe these reward
funcions and their fitness, try to write a reward function to further improve the fitness.
(Reward)
```

## B  DISCUSSION

RoboMoRe is designed to be task-agnostic and broadly generalizable, yet several directions remain to strengthen its practicality and scope. First, we will expand evaluation to more demanding benchmarks, especially soft-robotics platforms Graule et al. (2022); Gazzola et al. (2018); Zhang et al. (2019), where single-robot training can exceed a day and makes large-scale search nontrivial. Second, incorporating vision-language models could improve generalization and reduce prompt engineering by injecting visual priors and design commonsense.

A core next step is transferring RoboMoRe from simulation to hardware. Concretely, we will: (i) instantiate co-designed morphologies on modular robot platforms, mapping MJCF parameters to URDF and controller settings; (ii) use domain randomization and hardware-in-the-loop evaluation to close the sim-to-real gap; (iii) assess energy efficiency, robustness, and safety under real disturbances; and (iv) study bio-inspired embodiments where joint optimization of morphology and reward can yield more efficient locomotion and manipulation. Together, these efforts aim to validate RoboMoRe on physical systems while preserving its sample efficiency and diversity benefits.

## C  LIMITATION

**Limited Evaluation Loop**   Our study is restricted to tasks defined on parameterized models (*e.g.*, MuJoCo XML descriptions with adjustable morphology parameters or structure matrix in soft voxel robots). While this setting allows controlled evaluation and systematic optimization, it does not fully capture the challenges of unstructured design spaces or real-world hardware constraints such as CAD models.

**Simulation-to-Reality Gap**   Our framework has so far been evaluated only in virtual environments. While simulation provides a controlled and efficient testbed, it inevitably simplifies real-world conditions such as sensor noise, actuation limits, material wear, and unmodeled dynamics. As a result, the effectiveness of RoboMoRe on physical robots remains unverified, and its direct transferability to hardware is still uncertain.

**Lack of In-depth Optimization Theory**   While our empirical results demonstrate the effectiveness of RoboMoRe, we do not provide rigorous theoretical analysis of its optimization properties, such as convergence guarantees, sample complexity, or diversity–quality trade-offs.

**Prompt Specification Dependence**   Although our prompting is designed to be a relatively-general scheme, it still requires lightweight specification of design parameters in output format (semantics, valid ranges, and basic constraints). Empirically, this dependence becomes stronger as tasks grow in parameter count and coupling complexity. While this requirement can be reduced by adding an automated parameter interpreter (*e.g.*, parsing MJCF/URDF to infer names, ranges, and constraints) or by letting the LLM first self-interpret parameters before generation, we leave fully removing this dependence to future work and anticipate it will diminish as model capabilities improve.

## D  MOTIVATION

To verify our hypothesis, we conducted experiments to quantitatively evaluate the performance of two representative robot morphologies, short leg ant and long leg ant. By designing different reward functions, we encouraged each robot to adopt distinct motion modalities. Specifically, we encouraged the robot to adopt a rolling motion along the ground by augmenting the template reward function with an angular velocity term and a jumping motion on the air by augmenting the template reward function with a bouncing reward term.

As shown in Fig. 2, the experimental results indicate that the short leg ant, characterized by a bulky body and low center of gravity, exhibits low fitness in the jumping modality but achieves significantly higher fitness when utilizing the rolling modality. Conversely, the long leg ant robot, benefiting from its slender limbs and elevated center of gravity, performs exceptionally well in the jumping modality. However, its motion efficiency drastically decreases under the rolling modality due to frequent overturning and difficulty recovering balance. These results clearly illustrate the strong dependence

of robot performance on appropriate motion modalities, highlighting that optimal efficiency can only be achieved when morphology and modality are suitably matched.

In addition, Table 7 presents the fitness and efficiency of the ant robot under different conditions. We observe that the long-leg ant exhibits a clear advantage in terms of efficiency, but only when paired with a well-suited reward function. This finding underscores the importance of co-optimizing both morphology and control strategy in robot design to achieve optimal efficiency.

Table 7: **Fitness and Efficiency on four cases.**

| Case | (a) | (b) | (c) | (d) |
|------|------|------|------|------|
| Fitness | 108.83 | 37.39 | 59.94 | 5.13 |
| Efficiency | 288.76 | 109.50 | 1457.65 | 124.75 |

## E    ENVIRONMENT AND TASK DETAILS

This section provides additional details on the environments and tasks used in our experiments. We build upon standard environments from MuJoCo Gym, including Ant, Walker, Hopper, Half-Cheetah, and Swimmer, and use their built-in reward functions and morphologies as human-designed baselines for comparison. To further evaluate the robustness of RoboMoRe across diverse settings, we introduce three customized environments: Ant-Desert, Ant-Jump, and Ant-Powered. Task descriptions are used as input to both the Morphology Design Prompt and the Reward Shaping Prompt. The output format constrains the LLM's response to a set of robot morphology parameters, which are then fed into our custom design scripts to generate a complete MuJoCo XML file, following Ha (2019). We will open-source these design scripts alongside the code.

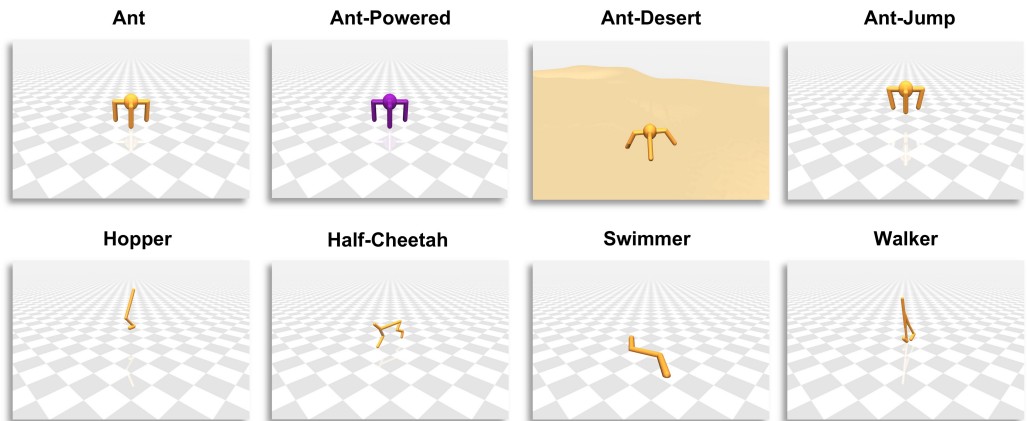

Figure 9: **Human designed agents in eight different environments for visualization.** Purple indicates agents in an augmented gear power.

## E.1 ANT

### E.1.1 TASK DESCRIPTION

```
The ant is a 3D quadruped robot consisting of a torso (free rotational body) with four legs
attached to
it, where each leg has two body parts. The goal is to coordinate the four legs to move in the
forward
(right) direction by applying torque to the eight hinges connecting the two body parts of
each leg and
the torso (nine body parts and eight hinges).
```

### E.1.2 OUTPUT FORMAT

```
Please output in json format without any notes:
{ "parameters": [<param1>, <param2>, ..., <param10>],
"description": "<your simple design style description>", }
#   param1 is the size of the torso, positive.
#   param2 is is the end attachment x point of leg, positive.
#   param3 is is the eend attachment y point of leg, positive.
#   param4 is is the end attachment x point of hip, positive.
#   param5 is the end attachment y point of hip, positive.
#   param6 is the end attachment x point of ankle, positive.
#   param7 is the end attachment y point of ankle, positive.
#   param8 is the size of the leg, positive.
#   param9 is the size of the hip, positive.
#   param10 is the size of the ankle, positive.
```

## E.2 ANT-POWERED

### E.2.1 TASK DESCRIPTION

```
The ant is a 3D quadruped robot consisting of a torso (free rotational body) with four legs
attached to it, where each leg has two body parts. The goal is to coordinate the four legs to
move in the forward (right) direction by applying torque to the eight hinges connecting the
two body parts of each leg and the torso (nine body parts and eight hinges)
```

### E.2.2 OUTPUT FORMAT

```
Please output in json format without any notes:
{ "parameters": [<param1>, <param2>, ..., <param10>],
"description": "<your simple design style description>", }
#   param1 is the size of the torso, positive.
#   param2 is is the end attachment x point of leg, positive.
#   param3 is is the end attachment y point of leg, positive.
#   param4 is is the end attachment x point of hip, positive.
#   param5 is the end attachment y point of hip, positive.
#   param6 is the end attachment x point of ankle, positive.
#   param7 is the end attachment y point of ankle, positive.
#   param8 is the size of the leg, positive.
#   param9 is the size of the hip, positive.
#   param10 is the size of the ankle, positive.
```

## E.3 ANT-DESERT

### E.3.1 TASK DESCRIPTION

```
The ant is a 3D quadruped robot consisting of a torso (free rotational body) with four legs
attached to it, where each leg has two body parts. The goal is to coordinate the four legs to
move in the forward (right) direction by applying torque to the eight hinges connecting the
two body parts of each leg and the torso (nine body parts and eight hinges).
```

### E.3.2 OUTPUT FORMAT

```
Please output in json format without any notes:
{ "parameters": [<param1>, <param2>, ..., <param10>],
"description": "<your simple design style description>", }
```

```
#   param1 is the size of the torso, positive.
#   param2 is is the end attachment x point of leg, positive.
#   param3 is is the end attachment y point of leg, positive.
#   param4 is is the end attachment x point of hip, positive.
#   param5 is the end attachment y point of hip, positive.
#   param6 is the end attachment x point of ankle, positive.
#   param7 is the end attachment y point of ankle, positive.
#   param8 is the size of the leg, positive.
#   param9 is the size of the hip, positive.
#   param10 is the size of the ankle, positive.
```

### E.4  ANT-JUMP

#### E.4.1  TASK DESCRIPTION

```
The ant is a 3D quadruped robot consisting of a torso (free rotational body) with four legs
attached to it, where each leg has two body parts. The goal is to coordinate the four legs to
jump in the upward direction by applying torque to the eight hinges connecting the two body
parts of each leg and the torso (nine body parts and eight hinges)
```

#### E.4.2  OUTPUT FORMAT

```
Please output in json format without any notes:
{ "parameters": [<param1>, <param2>, ..., <param10>],
"description": "<your simple design style description>", }
#   param1 is the size of the torso, positive.
#   param2 is is the end attachment x point of leg, positive.
#   param3 is is the end attachment y point of leg, positive.
#   param4 is is the end attachment x point of hip, positive.
#   param5 is the end attachment y point of hip, positive.
#   param6 is the end attachment x point of ankle, positive.
#   param7 is the end attachment y point of ankle, positive.
#   param8 is the size of the leg, positive.
#   param9 is the size of the hip, positive.
#   param10 is the size of the ankle, positive.
```

### E.5  HOPPER

#### E.5.1  TASK DESCRIPTION

```
The hopper is a two-dimensional one-legged figure consisting of four main body parts – the
torso at the top, the thigh in the middle, the leg at the bottom, and a single foot on which
the entire body rests. The goal is to make hops that move in the forward (right) direction by
applying torque to the three hinges that connect the four body parts.
```

#### E.5.2  OUTPUT FORMAT

```
Please output in json format without any notes:
{ "parameters": [<param1>, <param2>, ..., <param10>],
"description": "<your simple design style description>", }
#   param1 is the size of the torso, positive.
#   param2 is is the end attachment x point of leg, positive.
#   param3 is is the end attachment y point of leg, positive.
#   param4 is is the end attachment x point of hip, positive.
#   param5 is the end attachment y point of hip, positive.
#   param6 is the end attachment x point of ankle, positive.
#   param7 is the end attachment y point of ankle, positive.
#   param8 is the size of the leg, positive.
#   param9 is the size of the hip, positive.
#   param10 is the size of the ankle, positive.
```

### E.6  HALF-CHEETAH

```
The HalfCheetah is a 2-dimensional robot consisting of 9 body parts and 8 joints connecting
them (including two paws). The goal is to apply torque to the joints to make the cheetah run
forward (right) as fast as possible, with a positive reward based on the distance moved
forward and a negative reward for moving backward.
```

```
The cheetah's torso and head are fixed, and torque can only be applied to the other 6 joints
over the front and back thighs (which connect to the torso), the shins (which connect to the
thighs), and the feet (which connect to the shins), and the goal is to move as fast as
possible towards the right by applying torque to the rotors and using fluid friction.
```

### E.6.1   OUTPUT FORMAT

```
Please output in json format without any notes:
{"parameters": [<param1>, <param2>, ..., <param24>],
"description": "<your simple design style decription>",}
Parameters Description:
# param1: start point of torso (x), negtive
# param2: end point of torso (x) and start point of head (x), positive
# param3: end point of head (x).
# param4: end point of head (z).
# back leg parameters
# param5: end point of bthigh (x) and start point of bshin (x), sometimes positive.
# param6: end point of bthigh (z) and start point of bshin (z), negative.
# param7: end point of bshin (x) and start point of bfoot (x), sometimes negtive.
# param8: end point of bshin (z) and start point of bfoot (z), negative.
# param9: end point of bfoot (x), sometimes positive.
# param10: end point of bfoot (z), negative.
# forward leg parameters
# param11: end point of fthigh (x) and start point of fshin (x), sometimes negative.
# param12: end point of fthigh (z) and start point of fshin (z), negative.
# param13: end point of fshin (x) and start point of ffoot (x), sometimes positive.
# param14: end point of fshin (z) and start point of ffoot (z), negative.
# param15: end point of ffoot (x), sometimes positive.
# param16: end point of ffoot (z), negative.
# size information
# param17: torso capsule size.
# param18: head capsule size.
# param19: bthigh capsule size.
# param20: bshin capsule size.
# param21: bfoot capsule size.
# param22: fthigh capsule size.
# param23: fshin capsule size.
# param24: ffoot capsule size.
```

## E.7   SWIMMER

```
The swimmers consist of three or more segments ('links') and one less articulation joints
('rotors') - one rotor joint connects exactly two links to form a linear chain. The swimmer
is suspended in a two-dimensional pool and always starts in the same position (subject to
some deviation drawn from a uniform distribution), and the goal is to move as fast as
possible towards the right by applying torque to the rotors and using fluid friction.
```

### E.7.1   OUTPUT FORMAT

```
Please output in json format without any notes:
{"parameters": [<param1>, <param2>, ..., <param6>],
"description": "<your simple design style decription>",}
Parameters Description:
#   param1 is the length of first segment, positive.
#   param2 is the length of second segment, positive.
#   param3 is the length of third segment, positive.
#   param4 is the size of first segment, positive.
#   param5 is the size of second segment, positive.
#   param6 is the size of third segment, positive.
```

## E.8   WALKER

### E.8.1   TASK DESCRIPTION

```
The walker is a two-dimensional bipedal robot consisting of seven main body parts - a single
torso at the top (with the two legs splitting after the torso), two thighs in the middle
below the torso, two legs below the thighs, and two feet attached to the legs on which the
entire body rests. The goal is to walk in the forward (right) direction by applying torque to
the six hinges connecting the seven body parts.
```

### E.8.2 OUTPUT FORMAT

```
Please output in json format without any notes:
{ "parameters": [<param1>, <param2>, ..., <param10>],
"description": "<your simple design style description>", }
#   param1 is the size of the torso, positive.
#   param2 is is the end attachment x point of leg, positive.
#   param3 is is the end attachment y point of leg, positive.
#   param4 is is the end attachment x point of hip, positive.
#   param5 is the end attachment y point of hip, positive.
#   param6 is the end attachment x point of ankle, positive.
#   param7 is the end attachment y point of ankle, positive.
#   param8 is the size of the leg, positive.
#   param9 is the size of the hip, positive.
#   param10 is the size of the ankle, positive.
```

# F IMPLEMENTATION DETAILS

In line with standard reinforcement learning practices, we employ distributed trajectory sampling across multiple CPU threads to accelerate training. Each model is trained using four random seeds on a system equipped with one AMD EPYC 7T83 processor and a single NVIDIA RTX 4080 Super GPU. Our framework is built on Python 3.8.20, MuJoCo 2.3.1, Stable-Baselines3 2.4.1, and CUDA 12.4. For all environments considered, training a single policy for 5e5 steps takes approximately 15 minutes using

Table 8: **Coarse-to-Fine Optimization Hyperparameters**

| Hyperparameter | Value |
|---|---|
| $\mathcal{N}_\mathcal{M}$ | 25 |
| $\mathcal{N}_\mathcal{R}$ | 5 |
| $\mathcal{N}_{\mathcal{MS}}$ | 3 |
| $k$ | 5% |

24 CPU cores and one RTX 4080 Super GPU. GPT-4-turbo serves as the foundation model in most of our experiments. As shown in Table 8, for coarse optimization, we empirically set the number of morphologies $\mathcal{N}_\mathcal{M}$ to 25, the number of reward functions $\mathcal{N}_\mathcal{R}$ to 5, and the number of Morphology Screening $\mathcal{N}_{\mathcal{MS}}$ to 3, for trade-off between speed and design quality. For fine optimization, we select $k$ as 5% best candidates from coarse stage. This remains computationally efficient and practical—especially when compared to conventional methods like Bayesian Optimization or Evolutionary Algorithms, which often require several thousand iterations to converge.

**Policy Training** We use the Soft Actor-Critic (SAC) algorithm Haarnoja et al. (2018) from Stable-Baselines3 as the reinforcement learning backbone for all experiments, ensuring consistency across tasks. Hyperparameters are configured based on the recommendations from Gym SpinningUp Achiam (2018), which are well-suited for training both human-designed robot morphologies and their corresponding reward functions. To accelerate training, we leverage parallel environments via SubprocVecEnv with 16 instances. Experimental results are averaged over 100 independent runs to mitigate randomness. Each candidate is initially trained for $5 \times 10^5$ steps for efficiency, and the top-performing candidate is retrained for $10^6$ steps to ensure a fair comparison with alternative methods such as Bayesian Optimization (BO) and Eureka—both of which typically converge within this training budget across most tasks Achiam (2018). Table 9 demonstrates detailed parameters for policy training.

Table 9: **SAC Training Hyperparameters**

| Parameter | Value |
|---|---|
| Number of environments | 16 |
| Learning rate | 3e-4 |
| Buffer size | 2,000,000 |
| Learning starts | 10,000 |
| Batch size | 1024 |
| $\tau$ | 0.005 |
| $\gamma$ | 0.99 |
| Train frequency | 8 |
| Gradient steps | 4 |
| Policy kwargs | [512, 512] |

**Comparison Methods** We adopt Bayesian Optimization with a batch size of 1 and 100 iterations, using a Gaussian Process surrogate model with a Matérn 5/2 kernel, automatic relevance determination (ARD), and the Expected Improvement (EI) acquisition function. For Eureka, we follow the recommended configuration with 5 iterations and 16 populations per iteration.

# G  ADDITIONAL RESULTS

## G.1  IMPLEMENTATION ON MANIPULATION TASKS

We adapted the task descriptions from the DM-Control paper to our prompt design and developed a custom script to bridge the DM-Control Suite with OpenAI Gym. We evaluated each design using three metrics: (a) Fitness, defined as the speed of successful manipulation; (b) Efficiency, computed as fitness normalized by robot volume; and (c) Success rate, measured as the percentage of successful completions. Each design was independently tested over 100 trials and the results were averaged. All other experimental settings followed those in the main RoboMoRe framework. It should be noted that, for this task, we provide the LLM with human template morphology parameters, but we do not include any descriptions of the output format parameters.

## G.2  IMPLEMENTATION ON FREE-FORM DESIGN

We further evaluate RoboMoRe on EvoGym, a benchmark for free-form soft voxel robot (SVR) design. The framework was applied without introducing task-specific prompts or context learning. Task descriptions and output format were directly taken from LASeR main paper, and the output format was adapted to match the SVR specification. Only minor adjustments were made to the training setup, such as increasing batch size and the number of parallel environments.

Table 10 summarizes the results across four EvoGym tasks. RoboMoRe achieves competitive or superior performance relative to Bayesian Optimization (BO) and Genetic Algorithms (GA). For example, RoboMoRe reaches 3.12 on Pusher-v0, compared to 2.74 for BO and 2.15 for GA. On Jumper-v0, RoboMoRe maintains stable positive performance (0.077), whereas both baselines produce negative results. On average across tasks, RoboMoRe achieves 2.01, outperforming BO (1.81) and GA (0.82). These findings indicate that RoboMoRe can be directly extended to free-form robot co-design without requiring extensive manual reconfiguration, highlighting its generality and adaptability.

Table 10: Performance on EvoGym free-form soft voxel robot (SVR) tasks. RoboMoRe is compared against Bayesian Optimization (BO) and Genetic Algorithm (GA).

| Method | Walker-v0 | Carrier-v0 | Pusher-v0 | Jumper-v0 | Average |
|---|---|---|---|---|---|
| RoboMoRe | 3.91 | **0.93** | **3.12** | **0.08** | **2.01** |
| BO | **3.96** | 0.79 | 2.74 | -0.24 | 1.81 |
| GA | 2.37 | 0.36 | 2.15 | -1.61 | 0.82 |

## G.3  ADDITIONAL ANALYSIS: DIVERSITY REFLECTION VS. LASeR DIRECT

To further investigate this distinction, we conducted an additional experiment comparing RoboMoRe with the LASeR DiRect mechanism, using the implementation provided in the official LASeR github repository LASeR (2024). The EvoGym Walker-v0 environment was adopted as the benchmark, and twenty robot designs were generated for each method. Structural diversity was quantified using two complementary measures. The first is Cosine Diversity, which captures the directional differences between robot structures in a high-dimensional embedding space. The second is Hamming Diversity, which measures voxel-level differences in occupied positions across robot designs.

Table 11: Comparison of RoboMoRe and LASeR on structural diversity in EvoGym Walker-v0.

| Method | Cosine Diversity ↑ | Hamming Diversity ↑ | Tokens ↓ | Time (s) ↓ |
|---|---|---|---|---|
| LASeR | 0.3189 | 16.79 | 54,807 | 315.4 |
| **RoboMoRe** | **0.7187** | **37.83** | **28,537** | **57.8** |

The results show that RoboMoRe achieves higher structural diversity on both metrics while requiring fewer tokens and significantly less generation time. These findings support the claim that RoboMoRe's

diversity reflection mechanism is not only more general but also more efficient, extending beyond SVR-specific applications.

### G.3.1 ANALYSIS ON DIVERSITY REFLECTION AND MORPHOLOGY SCREENING

To more concretely demonstrate the effectiveness of the Diversity Reflection mechanism, we visualize the $5 \times 25$ efficiency matrices across multiple tasks for Random Sample, RoboMoRe w/o Diversity Reflection (DR), and RoboMoRe. Due to the large variation in data magnitude, we apply a logarithmic scale to the color map. As shown in Fig. 10, robots generated by Random Sample generally exhibit low efficiency, whereas those designed by RoboMoRe consistently achieve significantly higher performance across most tasks (see Table 4). Notably, while RoboMoRe w/o DR still achieves reasonably good results, the lack of morphological diversity causes the selected samples to concentrate around local optima, thereby leading to a slightly lower average efficiency compared to the full RoboMoRe configuration.

Fig. 10 further illustrates the rationale behind *Morphology Screening*: morphology largely determines the upper bound of co-optimization performance. Morphologies that perform poorly under the initial general reward consistently exhibit low performance across other reward functions as well. This observation indicates that it is unnecessary to optimize all reward functions for low-performing morphologies, and comparable results can be achieved by focusing only on promising candidates.

Table 12: Efficiency Comparison between RoboMoRe (w/o Morphology Screening) and (w/ Morphology Screening).

| Method | Ant | Walker | Swimmer | Half-Cheetah | Hopper | Ant-Powered | Ant-Jump | Ant-Desert |
|---|---|---|---|---|---|---|---|---|
| w/o Morphology Screening (25×5) | 10464.17 | 1482.59 | 21931.14 | 129440.98 | 1951.44 | 2777.38 | 396.45 | 8001.67 |
| w/ Morphology Screening (37) | 10464.17 | 1482.59 | 21931.14 | 129440.98 | 1701.61 | 2777.38 | 396.45 | 8001.67 |

Analysis of Moprhology Screening is also shown here. Rather than exhaustively evaluating all $25 \times 5$ = 125 reward-morphology pairs, RoboMoRe first evaluates all 25 morphologies with a fixed reward (e.g., reward0), then selects the top-3 morphologies to evaluate across the remaining 4 rewards. This results in only 37 evaluations ($25 + 3 \times 4$) while still recovering the same global optimum as shown in Table 12.

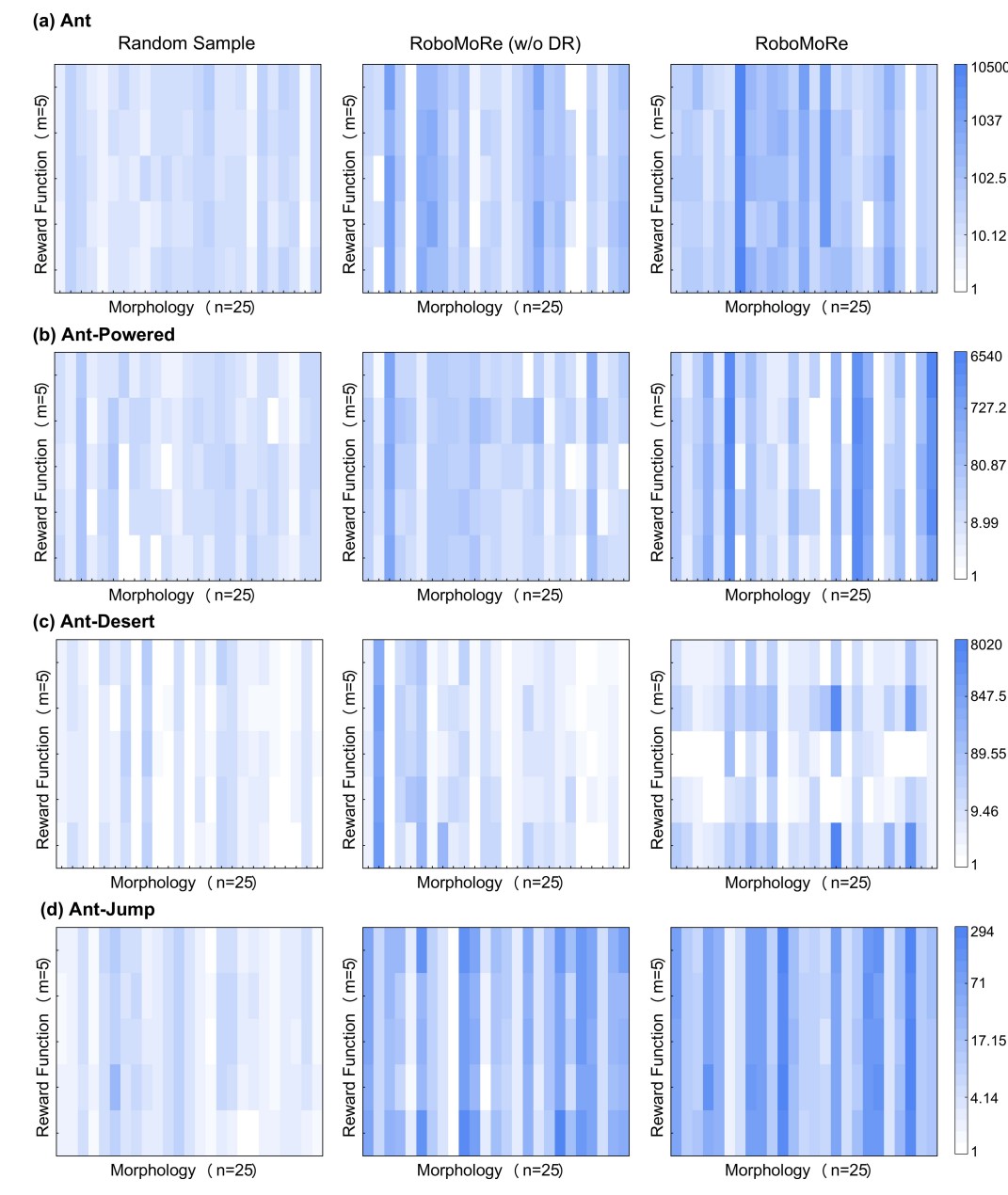

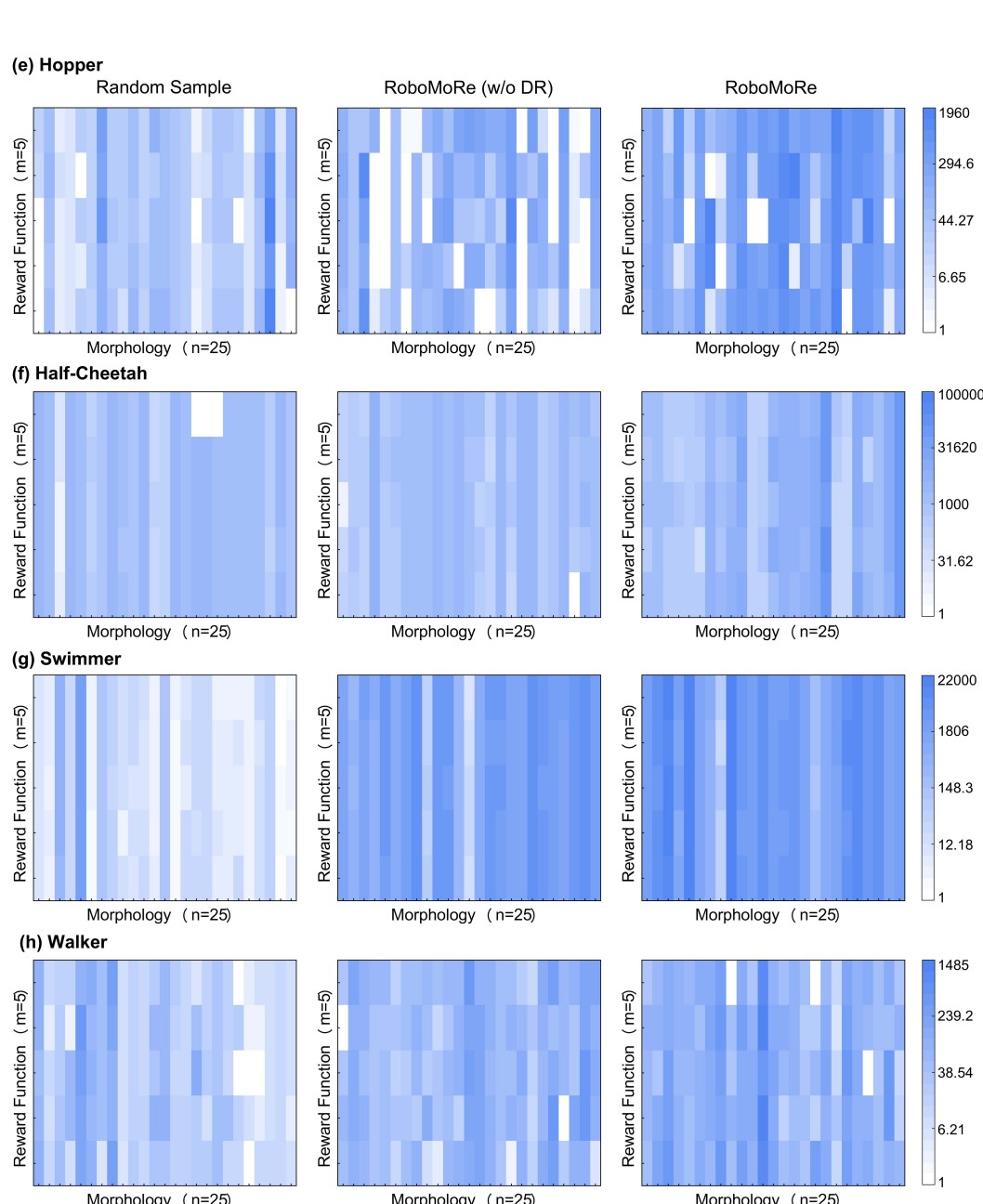

Figure 10: **Coarse optimization results across 8 tasks (a–h) for Random Sample, RoboMoRe w/o Diversity Reflection (DR), and RoboMoRe.**

### G.4 ROBUSTNESS ACROSS DIFFERENT GEAR POWERS

To validate the robustness of RoboMoRe-generated designs across varying levels of motor power, we selected the top-performing design from the Ant-Powered task and retrained the model under five additional power settings. These power levels were configured directly through the MuJoCo XML files. The results, summarized in Fig. 11, show that the design discovered by RoboMoRe demonstrates strong adaptability and maintains consistent performance across a wide range of actuation capabilities.

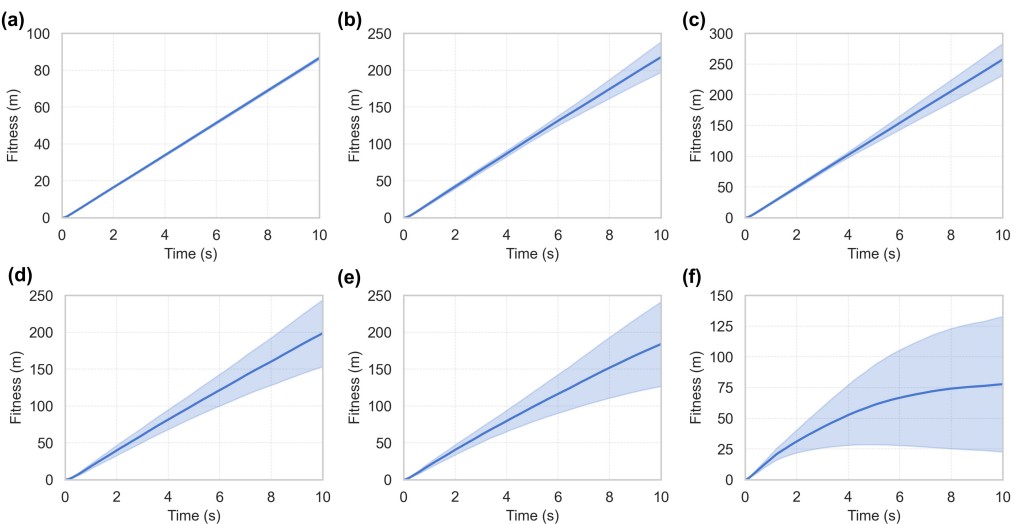

Figure 11: **Different power settings and corresponding results.** (a) Ant-Powered-50. (b) Ant-Powered-100. (c) Ant-Powered-150. (d) Ant-Powered-200. (e) Ant-Powered-250. (f) Ant-Powered-300.

### G.5 ROBUSTNESS ACROSS DIFFERENT TERRAINS

To further validate the robustness of RoboMoRe-generated designs across diverse terrains, we selected the top-performing design in Ant-Desert task and retrained the model on additional three distinct environments: ground, snow, and hills. These terrains were procedurally generated using Perlin noise and calibrated with realistic friction coefficients to reflect natural surface properties. The results, summarized in Fig. 12, demonstrate that the design discovered by RoboMoRe exhibits strong adaptability and consistent performance across all tested terrains, highlighting its robustness in diverse physical conditions.

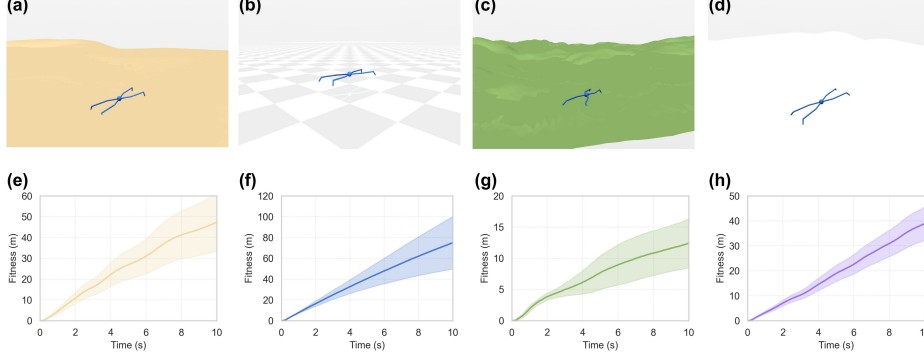

Figure 12: **Different terrain environments and corresponding results.** (a) Ant-Desert. (b) Ant-Ground. (c) Ant-Hills. (d) Ant-Snow. (e) Results for Ant-Desert. (f) Results for Ant-Ground. (g) Results for Ant-Hills. (h) Results for Ant-Snow.

# H ALGORITHM DETAILS

---

**Algorithm 1: Coarse-to-Fine Optimization**

---

**Data:** Number of morphologies $\mathcal{N}_{\mathcal{M}}$, number of rewards $\mathcal{N}_{\mathcal{R}}$, number of morphology screening candidates $\mathcal{N}_{\mathcal{MS}}$;
Morphology prompt $P_{\mathcal{M}}$, reward prompt $P_{\mathcal{R}}$, diversity prompt $P_{\text{diversity}}$;
RL training algorithm $\text{RL}(\cdot)$ (*e.g.*, SAC); Fine optimization selection count $k$;
**Result:** Optimized morphology $\theta^*$ and reward function $r^*$

*// Coarse Optimization: Grid Search with Diversity Reflection.*;
$\Theta \leftarrow \emptyset$;
$\theta_1 \leftarrow \text{ProposeMorphology}(P_{\mathcal{M}})$;
$\Theta \leftarrow \Theta \cup \{\theta_1\}$;
**for** $i \leftarrow 2$ **to** $\mathcal{N}_{\mathcal{M}}$ **do**
$\quad \theta_i \leftarrow \text{ProposeMorphology}(\Theta, P_{\mathcal{M}}, P_{\text{diversity}})$;
$\quad \Theta \leftarrow \Theta \cup \{\theta_i\}$;
$\mathcal{R} \leftarrow \emptyset$;
$r_1 \leftarrow \text{ProposeReward}(P_{\mathcal{R}})$;
$\mathcal{R} \leftarrow \mathcal{R} \cup \{r_1\}$;
**for** $j \leftarrow 2$ **to** $\mathcal{N}_{\mathcal{R}}$ **do**
$\quad r_j \leftarrow \text{ProposeReward}(\mathcal{R}, P_{\mathcal{R}}, P_{\text{diversity}})$;
$\quad \mathcal{R} \leftarrow \mathcal{R} \cup \{r_j\}$;
$\mathcal{F} \leftarrow \emptyset$;
$\mathcal{F}, \Theta = Morphology\_Screen(\mathcal{F}, \Theta, \mathcal{N}_{\mathcal{MS}})$;
**foreach** $\theta \in \Theta$ **do**
$\quad$ **foreach** $r \in \mathcal{R}$ **do**
$\quad\quad F \leftarrow \text{RL}(\theta, r)$;
$\quad\quad \mathcal{F} \leftarrow \mathcal{F} \cup \{(\theta, r, F)\}$;

$\mathcal{S}_{\text{coarse-best}} \leftarrow \text{Top}(\mathcal{F}, k)$;
*// Fine Optimization: Alternating Morphology and Reward Optimization*;
$\mathcal{S}_{\text{fine}} \leftarrow \emptyset$;
**foreach** $(\theta, r) \in \mathcal{S}_{coarse\text{-}best}$ **do**
$\quad \theta^* \leftarrow \theta, r^* \leftarrow r$;
$\quad F^* \leftarrow \text{RL}(\theta^*, r^*)$;
$\quad$ **while** *not converged* **do**
$\quad\quad$ improved $\leftarrow$ False;
$\quad\quad$ *// Morphology Optimization*;
$\quad\quad \theta' \leftarrow \text{ImproveMorphology}(\theta^*, r^*, \Theta, \mathcal{F})$;
$\quad\quad F' \leftarrow \text{RL}(\theta', r^*)$;
$\quad\quad$ **if** $F' > F^*$ **then**
$\quad\quad\quad \theta^* \leftarrow \theta', F^* \leftarrow F'$;;
$\quad\quad\quad$ improved $\leftarrow$ True;;
$\quad\quad$ *// Reward Function Optimization*;
$\quad\quad r' \leftarrow \text{ImproveReward}(\theta^*, r^*, \mathcal{R}, \mathcal{F})$;
$\quad\quad F' \leftarrow \text{RL}(\theta^*, r')$;
$\quad\quad$ **if** $F' > F^*$ **then**
$\quad\quad\quad r^* \leftarrow r', F^* \leftarrow F'$;;
$\quad\quad\quad$ improved $\leftarrow$ True;;
$\quad\quad$ **if** *not improved* **then**
$\quad\quad\quad$ break;
$\quad \mathcal{S}_{\text{fine}} \leftarrow \mathcal{S}_{\text{fine}} \cup \{(\theta^*, r^*, F^*)\}$;
**return** Best $(\theta^*, r^*)$ from $\mathcal{S}_{\text{fine}}$;

## I  PROOF OF TEMPERATURE SCALING ON DIVERSITY

To examine whether temperature scaling can improve diversity, we conducted a controlled experiment using the Ant task. Both morphology and reward diversity were evaluated under different temperature settings, and the results were compared against RoboMoRe's Diversity Reflection mechanism.

Table 13: Impact of temperature scaling on diversity. Morphology and reward diversity are averaged over 50 samples; motion modality diversity is assessed qualitatively.

| Method (Ant task) | Morphology Diversity ↑ | Reward Diversity ↓ | Motion Modality↑ |
|---|---|---|---|
| T=1.0 (Default) | 0.73 | 0.67 | Low |
| T=1.1 | 0.78 | 0.62 | Low |
| T=1.2 | 0.80 | 0.53 | Low |
| T=1.3 | 0.83 | 0.48 | Low |
| T=1.4 | 0.85 | 0.50 | Low (artifacts) |
| T=1.5 | 0.94 | 0.41 | Low (artifacts) |
| DR (T=1.0) | **0.87** | **0.45** | High |

The results from Fig. 13 indicate that temperature scaling can increase morphological diversity, particularly at higher values (*e.g.*, $T = 1.3$ and above). However, this comes with notable drawbacks. At elevated temperatures ($T \geq 1.4$), the LLM frequently produced malformed or syntactically invalid reward functions, and response latency increased significantly. Moreover, while temperature scaling enhances surface-level variability in reward outputs, it does not effectively promote behavioral diversity: motion modalities remained highly repetitive across trials.

By contrast, the Diversity Reflection mechanism explicitly considers the joint interaction between morphology and reward, leading to reward–morphology pairs that generate qualitatively distinct motion behaviors. This highlights its advantage over temperature-based sampling, which primarily induces superficial diversity at the cost of reliability.

**Conclusion.** Increasing a extremely high temperature (*e.g.*, T=1.5) can yield greater morphological variation, but it often degrades generation stability and fails to produce meaningful behavioral diversity. In comparison, Diversity Reflection provides a more robust and effective approach to achieving functional diversity in robot co-design.

## J  LLM USAGE STATEMENT

We acknowledge the use of large language models (LLMs) during the preparation of this work, LLMs were used to aid drafting and polish the writing.

## K  MORE VISUALIZATION RESULTS

### K.1  COMPARISON OF OPTIMAL MORPHOLOGY DESIGN VIA DIFFERENT METHODS

Fig. 13 presents the optimized morphology design. Evidently, RoboMoRe is capable of producing highly efficient structures, demonstrating its effectiveness in morphology optimization.

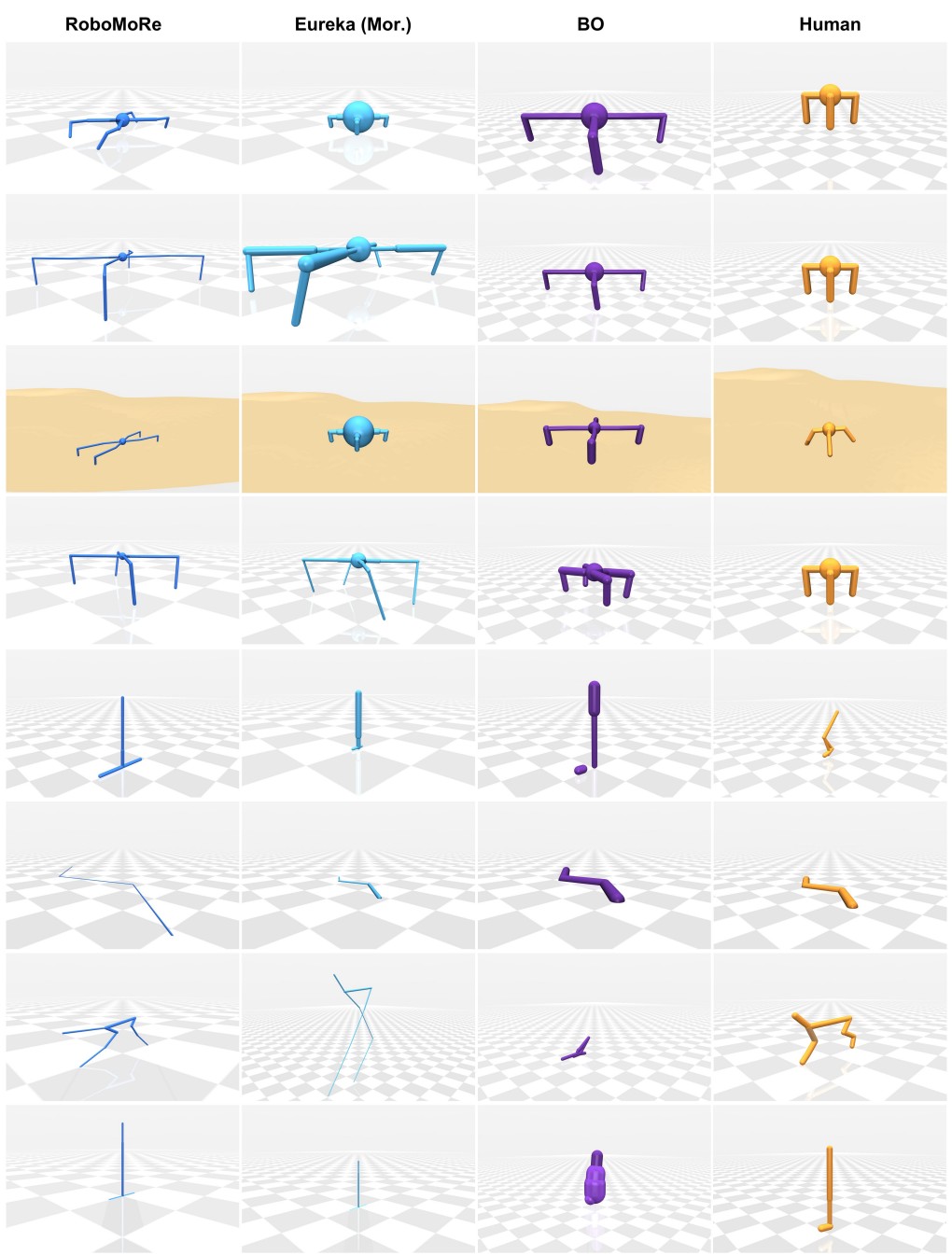

Figure 13: **Optimized morphology with comparison methods.**

## K.2  ANT: BEST REWARD FUNCTION

```python
import numpy as np
def _get_rew(self, x_velocity: float, action):
    # Encourage not just moving forward but also involving some controlled lateral movement
    # to demonstrate agility and dynamic control capability.

    # Reward for moving forward
    forward_reward = self._forward_reward_weight * x_velocity

    # Reward for controlled lateral movement (moderate y_velocity to demonstrate lateral agility)
    target_y_velocity = 0.5  # moderate lateral speed target, can be tuned
    lateral_movement_reward = -np.abs(self.data.qvel[1] - target_y_velocity) * self._forward_reward_weight

    # Minimize the control effort to promote energy efficiency.
    control_cost = self.control_cost(action)

    # Health reward for maintaining a physically feasible and stable posture.
    health_reward = self.healthy_reward

    # Total reward computation
    reward = forward_reward + lateral_movement_reward - control_cost + health_reward

    # Reward details for debugging and analysis purposes
    reward_info = {
        'forward_reward': forward_reward,
        'lateral_movement_reward': lateral_movement_reward,
        'control_cost': control_cost,
        'health_reward': health_reward
    }

    return reward, reward_info
```

## K.3  ANT-POWERED: BEST REWARD FUNCTION

```python
import numpy as np
def _get_rew(self, x_velocity: float, action):
    # Encourage forward movement with a target velocity for optimal speed
    target_velocity = 1.0  # Desired forward velocity
    velocity_error = np.abs(x_velocity - target_velocity)  # Calculate deviation from target
    forward_reward = self._forward_reward_weight * np.exp(-velocity_error)  # Exponential decay for reward

    # Penalize lateral (y-direction) movement to maintain focus on forward motion
    y_velocity_penalty = -abs(self.data.qvel[1])  # Penalize any motion in the y direction

    # Introduce a reward for efficient alternate leg movement to promote stability and locomotion style
    leg_pair_1 = np.abs(action[0] + action[1] - (action[4] + action[5]))
    leg_pair_2 = np.abs(action[2] + action[3] - (action[6] + action[7]))
    alternate_leg_movement_reward = (np.exp(-leg_pair_1) + np.exp(-leg_pair_2)) / 2.0

    # Maintain healthy posture by rewarding stability within the Z-limits
    health_reward = self.healthy_reward

    # Evaluate the efficiency of movement through control costs
    control_cost = self.control_cost(action)

    # Penalize excessive contact forces to minimize instability
    contact_cost = self.contact_cost

    # Calculate total reward by combining all components
    reward = forward_reward + y_velocity_penalty + alternate_leg_movement_reward - control_cost + health_reward - contact_cost

    # Reward info for monitoring individual components
    reward_info = {
        "forward_reward": forward_reward,
        "y_velocity_penalty": y_velocity_penalty,
        "alternate_leg_movement_reward": alternate_leg_movement_reward,
        "control_cost": control_cost,
        "contact_cost": contact_cost,
        "health_reward": health_reward,
    }

    return reward, reward_info
```

## K.4 ANT-DESERT: BEST REWARD FUNCTION

```python
import numpy as np
def _get_rew(self, x_velocity: float, action):
    # Reward for forward movement, emphasizing an exponential growth to encourage high speeds.
    forward_reward = self._forward_reward_weight * np.exp(x_velocity - 1)  # Exponential
growth for incentivizing high speeds

    # Encourage lateral stability: penalize the absolute value of lateral (y-axis) velocity
    y_velocity = self.data.qvel[1]
    lateral_stability_penalty = -np.abs(y_velocity)  # Negative as we want to minimize
lateral movement

    # Control cost to penalize excessive energy usage
    control_cost = self.control_cost(action)

    # Keep the robot in a healthy state
    healthy_reward = self.healthy_reward

    # Minimize contact costs to promote gentle contacts with the ground
    contact_cost = self.contact_cost

    # Total reward calculation combining all components
    reward = forward_reward + healthy_reward + lateral_stability_penalty - control_cost -
contact_cost

    # Detailed reward breakdown for analysis and debugging
    reward_info = {
        'forward_reward': forward_reward,
        'lateral_stability_penalty': lateral_stability_penalty,
        'control_cost': control_cost,
        'contact_cost': contact_cost,
        'healthy_reward': healthy_reward
    }

    return reward, reward_info
```

## K.5 ANT-JUMP: BEST REWARD FUNCTION

```python
import numpy as np
def _get_rew(self, x_velocity: float, action):
    # Decomposing the reward function
    jump_height = self.data.body(self._main_body).xpos[2]  # Z position gives the height

    # Reward for jumping higher
    height_reward = np.exp(jump_height - 1)  # exponential reward starting from height 1

    # Control cost to make sure the robot uses minimum effort to jump
    control_cost = self.control_cost(action)

    # Contact cost to penalize excessive force usage in contacts, promoting smooth jumping
    contact_cost = self.contact_cost

    # Component to support healthy posture
    healthy_posture_reward = self.healthy_reward

    # Combination of different components of the reward
    reward = (
        height_reward * self._forward_reward_weight
        - contact_cost
        - control_cost
        + healthy_posture_reward
    )

    # Reward info for better analysis and debugging
    reward_info = {
        "height_reward": height_reward,
        "control_cost": control_cost,
        "contact_cost": contact_cost,
        "healthy_posture_reward": healthy_posture_reward,
    }

    return reward, reward_info
```

### K.6 HOPPER: BEST REWARD FUNCTION

```python
import numpy as np
def _get_rew(self, x_velocity: float, action):
    # Encourage efficient forward movement by not just rewarding speed but also smooth
progression
    smoothness_reward = -np.sum(np.abs(np.diff(action)))  # Decrease reward for large
differences in consecutive actions

    # Reward forward velocity, with an exponential component to prioritize higher speeds
exponentially
    exponential_speed_reward = self._forward_reward_weight * np.exp(x_velocity) - 1  # Using
exp to exponentially prefer higher velocities, subtract 1 to center around zero for small
velocities

    # Penalty for using too much control input, which promotes efficiency
    control_penalty = self._ctrl_cost_weight * np.sum(np.square(action))

    # Healthy state reward, keeping the hopper upright and in a healthy range
    health_bonus = self.healthy_reward

    # Total reward calculation
    total_reward = exponential_speed_reward + smoothness_reward - control_penalty +
health_bonus

    # Tracking reward details for better understanding and debugging
    reward_info = {
        'smoothness_reward': smoothness_reward,
        'exponential_speed_reward': exponential_speed_reward,
        'control_penalty': control_penalty,
        'health_bonus': health_bonus,
        'total_reward': total_reward
    }

    return total_reward, reward_info
```

### K.7 HALF-CHEETAH: BEST REWARD FUNCTION

```python
import numpy as np
def _get_rew(self, x_velocity: float, action):
    # Reward for moving forward emphasizing higher speeds
    forward_reward = self._forward_reward_weight * x_velocity

    # Calculate control cost using the predefined method
    control_cost = self.control_cost(action)

    # Reward for energy efficiency: velocity per control effort
    efficiency = x_velocity / (control_cost + 1e-5)  # Avoid division by zero
    normalized_efficiency_reward = np.exp(efficiency) - 1  # Shifted by -1 to normalize
around 0

    # Calculate smoothness reward: penalize fluctuations in velocity
    if not hasattr(self, 'prev_velocity'):
        self.prev_velocity = x_velocity
    smoothness_penalty = -np.abs(x_velocity - self.prev_velocity)  # Penalize changes in
velocity
    self.prev_velocity = x_velocity  # Update the previous velocity for the next step
    smoothness_reward = np.exp(smoothness_penalty) - 1  # Normalize the smoothness reward

    # Action symmetry bonus: rewards symmetrical actions between limbs
    if len(action) % 2 == 0:
        left_actions = action[1::2]
        right_actions = action[0::2]
        symmetry_penalty = -np.sum(np.abs(left_actions - right_actions))
    else:
        symmetry_penalty = 0
    symmetry_reward = np.exp(symmetry_penalty) - 1

    # Combine all components to form the total reward
    total_reward = forward_reward - control_cost + normalized_efficiency_reward +
smoothness_reward + symmetry_reward

    # Reward info dictionary for debugging and analysis
    reward_info = {
        'forward_reward': forward_reward,
        'control_cost': control_cost,
        'normalized_efficiency_reward': normalized_efficiency_reward,
        'smoothness_reward': smoothness_reward,
```

```
1944          'symmetry_reward': symmetry_reward,
1945          'total_reward': total_reward
1946      }
1947
1948      return total_reward, reward_info
1949
```

## K.8 SWIMMER: BEST REWARD FUNCTION

```
import numpy as np
def _get_rew(self, x_velocity: float, action):
    # Define efficient swimming characteristics. Aim to maintain balance between forward
motion and control.

    # Base forward reward for moving right, placed higher for higher velocities
    forward_reward = self._forward_reward_weight * (x_velocity ** 2)  # Quadratic reward
amplifies strong velocities

    # Introduce a reward for stability: Encourage minimal yawing (side ways movements).
    stability_reward = -0.5 * (self.data.qpos[1] ** 2)  # Penalizing lateral position squared
to discourage side motion

    # Punishing excessive control inputs, to favor smooth swimming rather than jerky movements
    control_penalty = self.control_cost(action)

    # Incentivize maintaining a certain baseline velocity (e.g., cruising speed), with a soft
penalty for deviations
    ideal_velocity = 1.0
    velocity_penalty = 0.5 * ((x_velocity - ideal_velocity) ** 2)  # Penalizes big deviations
from ideal

    # Compute the total reward
    total_reward = forward_reward + stability_reward - control_penalty - velocity_penalty

    # Debugging and analysis information containing detailed components of the reward
    reward_info = {
        'forward_reward': forward_reward,
        'stability_reward': stability_reward,
        'control_penalty': control_penalty,
        'velocity_penalty': velocity_penalty,
    }

    return total_reward, reward_info
```

## K.9 WALKER: BEST REWARD FUNCTION

```
import numpy as np
def _get_rew(self, x_velocity: float, action):
    # Reward for forward motion, scaled exponentially to favor higher speeds but with
diminishing returns
    forward_reward = np.exp(0.3 * x_velocity) - 1

    # Penalty to discourage excessive use of actuator torques, promoting energy efficiency
    control_penalty = self.control_cost(action)

    # Bonus for maintaining a healthy mechanics of motion, which includes staying upright
    health_bonus = self.healthy_reward

    # Additional reward for synchronous and rhythmic leg movements
    # We can leverage the sine of the sum of relevant joint angles to promote a smooth cyclic
locomotion
    angles = self.data.qpos[2:7]  # Assuming indices 2-6 are joint angles
    rhythmic_movement_bonus = np.sum(np.sin(angles))

    # Compute total reward considering all the components
    reward = forward_reward + rhythmic_movement_bonus + health_bonus - control_penalty

    # Organize detailed reward components for diagnostic purposes
    reward_info = {
        'forward_reward': forward_reward,
        'rhythmic_movement_bonus': rhythmic_movement_bonus,
        'health_bonus': health_bonus,
        'control_penalty': control_penalty
    }

    return reward, reward_info
```

