# OpenReview forum: "RoboMoRe: LLM-based Robot Co-design via Joint Optimization of Morphology and Reward"
_ICLR.cc/2026/Conference — Submitted to ICLR 2026_

### Official Review · Reviewer_rj41 · 2025-10-27

**Soundness:** 4
**Presentation:** 4
**Contribution:** 3
**Rating:** 8
**Confidence:** 3

**Summary:**

This paper addresses a fundamental challenge in evolutionary Robotics, i.e., robot design automation, and proposes a novel perspective that evolves reward functions simultaneously in order to account for distinct motion modalities across morphologies. Specifically, the authors introduce a dual-stage strategy: a coarse stage where an LLM is prompted to propose diverse and promising morphology-reward pairs, and a fine stage where the solutions are further optimized. Extensive experiments demonstrate the effectiveness of the proposed approach, RoboMoRe, compared with several baselines.

**Strengths:**

1.The paper is well-written and easy to follow. The abundant illustrations and qualitative results greatly help with reader’s understanding.

2.The paper is well motivated. The design of reward function is left simple and static in previous robot co-design studies. The introduction of reward shaping facilitates more customized and precise fitness evaluation across different morphologies, leading to improved optimization efficiency and overall performances.

3.I greatly appreciate the authors’ endeavour to examine their method in various settings, such as template-based and free-form robot design, as well as different environmental perturbations. The results validate their adaptability and generality across diverse scenarios.

4.The authors provide a comprehensive discussion of limitations and future directions, which would greatly inspire further studies.

**Weaknesses:**

1.I believe the motivating example given in Appendix D is worth incorporating into the main text, so that the logic flows more naturally.

2.The authors could also include a brief outline at the beginning of the appendices so that the contents are more organized.

**Questions:**

1.Since the reward functions are evolved by an LLM, I wonder whether this would implicitly introduce a bias towards robot morphologies with more interpretable motion patterns instead of those with high potential but less intuitive reward functions to evolve?

2.Could the authors provide a couple of example reward functions for soft voxel robots? Since the dynamics of SVRs are far less intuitive than articulated rigid robots, I wonder whether LLMs can still evolve interpretable reward functions?

---

> ### Author Response · Authors · 2025-11-21
> **Response to W1 & W2**
>
> **We sincerely thank the reviewer for the thoughtful and encouraging assessment of our work.** We truly appreciate your constructive feedback and the recognition of our contribution. Your comments are very helpful for improving the clarity and presentation of the paper!
>
> Below we address your questions and suggestions in detail:
>
> **W1.** I believe the motivating example given in Appendix D is worth incorporating into the main text, so that the logic flows more naturally.
>
> **Re:** Good idea! We have incorporated the motivating example from Appendix D into the main text to improve the logical flow.
>
> **W2.** The authors could also include a brief outline at the beginning of the appendices so that the contents are more organized.
>
> **Re:** Thank you for the helpful suggestion! We agree that an outline improves the readability of the appendices. We have now added a brief organizational overview at the beginning of the appendix section to make the structure clearer. Please see our newest version for details.

---

> > ### Author Response · Authors · 2025-11-21
> >
> > **Q1.** Since the reward functions are evolved by an LLM, I wonder whether this would implicitly introduce a bias towards robot morphologies with more interpretable motion patterns instead of those with high potential but less intuitive reward functions to evolve?
> >
> > **Re:** Thank you for the question. In RoboMoRe, LLM-generated reward functions do *not* bias toward “interpretable” behaviors, because rewards are **never used for evaluation**—they are only used to *train* controllers. All morphology–reward pairs are ultimately judged using the same **reward-independent fitness metric** (e.g., forward distance or jump height), so any morphology with high physical potential will outperform less effective ones regardless of how “intuitive” or “interpretable” its reward appears.
> >
> > In practice, we observe (*e.g.*, Figure 7) that the LLM routinely discovers non-intuitive motion patterns such as rolling, side-flipping, and crab-walking, indicating that the system does not favor only human-interpretable behaviors. High-performing designs emerge purely based on **environmental fitness**, not reward readability.
> >
> >
> > **Q2.** a couple of example reward functions for soft voxel robots.
> >
> > **Re:** Thank you for the question! Yes, in the SVR environments, the LLM is still able to generate valid and interpretable reward functions for soft voxel robots as well. The reward prompts are identical to those used for articulated robots, and the LLM produces simple, physically meaningful terms such as forward displacement, deformation penalties, and symmetry or oscillation incentives.
> >
> > For clarity, we provide three representative RoboMoRe-generated reward functions for SVRs. These examples are concise and follow the same structure as those shown for rigid-body robots, demonstrating that the method extends naturally to soft-voxel dynamics.
> >
> > ```jsx
> > # Sideways Hopping (lateral-impulse locomotion)
> >
> > import numpy as np
> > def _get_rew(self, x_velocity: float, action):
> >     # Forward progress (core task objective)
> >     forward_reward = self._forward_reward_weight * x_velocity
> >
> >     # Encourage rhythmic lateral impulses (side hops)
> >     y_vel = self.data.qvel[1]
> >     lateral_oscillation = np.sin(3 * self.data.time)         # target lateral rhythm
> >     lateral_reward = -np.abs(y_vel - lateral_oscillation)    # closer to the target is better
> >
> >     # Promote strong vertical impulses for hopping
> >     hop_velocity = self.data.qvel[2]
> >     hop_reward = np.maximum(0.0, hop_velocity)               # reward upward hopping motion
> >
> >     # Regularization terms
> >     control_cost = self.control_cost(action)
> >     contact_cost = self.contact_cost
> >     healthy_reward = self.healthy_reward
> >
> >     reward = (forward_reward
> >               + lateral_reward * 0.5
> >               + hop_reward * 0.3
> >               + healthy_reward
> >               - control_cost * 0.2
> >               - contact_cost * 0.1)
> >
> >     reward_info = dict(
> >         forward_reward=forward_reward,
> >         lateral_reward=lateral_reward,
> >         hop_reward=hop_reward,
> >         control_cost=control_cost,
> >         contact_cost=contact_cost,
> >         healthy_reward=healthy_reward
> >     )
> >     return reward, reward_info
> >
> > ```
> >
> > ```jsx
> > # Rolling Locomotion (low-center-of-gravity rolling)
> > import numpy as np
> > def _get_rew(self, x_velocity: float, action):
> >     forward_reward = self._forward_reward_weight * x_velocity
> >
> >     # Encourage rolling: high angular velocity around body axis (y or x axis depending on env)
> >     roll_rate = np.abs(self.data.qvel[3])   # body angular velocity
> >     rolling_reward = np.exp(roll_rate) - 1  # exponential boost for sustained rolling
> >
> >     # Encourage stable low posture typical of rolling morphologies
> >     height = self.data.body(self._main_body).xpos[2]
> >     low_posture_reward = np.exp(-(height - 0.25)**2)
> >
> >     control_cost = self.control_cost(action)
> >     contact_cost = self.contact_cost
> >     healthy_reward = self.healthy_reward
> >
> >     reward = (forward_reward
> >               + rolling_reward * 0.4
> >               + low_posture_reward * 0.2
> >               + healthy_reward
> >               - control_cost * 0.2
> >               - contact_cost * 0.1)
> >
> >     reward_info = dict(
> >         forward_reward=forward_reward,
> >         rolling_reward=rolling_reward,
> >         low_posture_reward=low_posture_reward,
> >         control_cost=control_cost,
> >         contact_cost=contact_cost,
> >         healthy_reward=healthy_reward
> >     )
> >     return reward, reward_info
> >
> > ```

---

> ### Author Response · Authors · 2025-11-27
>
> Hi Reviewer rj41, thank you again for your very encouraging and thoughtful review of our paper, as well as for the insightful questions and helpful presentation suggestions! We have implemented your recommendations and responded in detail to both questions regarding reward bias and soft voxel robot reward examples above. As the discussion period is nearing its end, we just wanted to kindly check whether there are any remaining questions or clarifications we could further address. We truly appreciate your time and valuable feedback—thank you for helping us improve the quality of the paper!

---

> > ### Comment · Reviewer_rj41 · 2025-11-28
> > **Response to rebuttal**
> >
> > I appreciate the authors' detailed response and will maintain my rating.

---

### Official Review · Reviewer_g8QY · 2025-10-31

**Soundness:** 4
**Presentation:** 3
**Contribution:** 3
**Rating:** 6
**Confidence:** 4

**Summary:**

The paper presents RoboMoRe, a framework that integrates large language models (LLMs) into the robot co-design process, enabling the joint optimization of morphology and control. By leveraging natural language descriptions and structured reasoning, the system aims to generate robot designs that are both functional and specialized for given tasks. The authors highlight how the LLM facilitates creative design exploration while remaining grounded through simulation-based evaluation. The approach is validated through multiple case studies showing task-specific robot designs and control policies optimized jointly via reinforcement learning and design iteration.

**Strengths:**

1. Insightful finding on co-design: The study convincingly shows that reward shaping must be robot-dependent, illustrating that task-based optimization alone fails to generalize across morphologies. This is an important and underexplored insight in robot design problem.

2. Timely and important topic: The intersection of LLMs and robotic design is a rapidly growing area with significant potential impact. The authors target an important problem - how generative models can extend the space of feasible robot designs.

3. Sound evaluation protocol: The experiments are thoughtfully designed, combining simulated environments with objective metrics that quantify both control performance and morphological diversity.

**Weaknesses:**

1. While the integration of LLMs is appealing, much of the technical pipeline (e.g., reinforcement learning for control, parametric morphology search) builds directly on established frameworks, so the question is whether the contribution is appealing enough to a general machine learning audience.

**Questions:**

1. How sensitive is the system to prompt phrasing? Did the authors test multiple prompt templates for reward generation, and if so, how consistent were the resulting performance outcomes?

---

> ### Author Response · Authors · 2025-11-21
> **Rebuttal**
>
> **We sincerely thank the reviewer for the positive and thoughtful assessment of our work!** Your feedback is very helpful, and we address the raised questions and concerns below.
>
> **W1.** While the integration of LLMs is appealing, much of the technical pipeline (*e.g.*, reinforcement learning for control, parametric morphology search) builds directly on established frameworks, so the question is whether the contribution is appealing enough to a general machine learning audience.
>
> **Re:** Thank you for raising this important question! While reinforcement learning and parametric morphology spaces are well-established, RoboMoRe’s contribution lies in introducing a new co-design formulation and a general algorithmic framework—Diversity Reflection, morphology screening, and alternating LLM-guided refinement—that enables joint optimization of morphology and reward in high-dimensional, non-differentiable spaces. These mechanisms significantly expand what *previous ML algorithms* can achieve in design optimization and demonstrate that LLMs can function as iterative optimizers rather than static generators. Therefore, although the training and simulation components rely on standard tools, they do not diminish the novelty or generality of the framework, which we believe is broadly relevant to the ML community.
>
> **Q1.** How sensitive is the system to prompt phrasing? Did the authors test multiple prompt templates for reward generation, and if so, how consistent were the resulting performance outcomes?
>
> **Re:** We emphasize that our prompts are intentionally kept minimal (please see Appendix A in our main paper), which helps ensure broad applicability without relying on task-specific heuristics. To further examine robustness under this minimal interface, we are very happy to additionally test four prompt variants that differ only in phrasing and structure:
>
> 1. *Original Prompt* – The default prompt used in our main experiments, with full task description + environment code + standard reward instructions.
> 2. *Minimal Prompt* – A highly condensed version that keeps only the essential instructions, used to test robustness under minimal guidance.
> 3. *Reordered Prompt (Constraints First)* – Same content as the original, but with all rules/constraints placed before the task description, used to test sensitivity to structural rearrangement.
> 4. *More Verbose Prompt* – A longer prompt with additional natural-language hints and clarifications, used to test whether extra descriptive context changes the outcome.
>
> For each prompt variant, we generated five reward functions, trained controllers on the Ant task (5×10⁵ steps), and report the best score. The four variants yield comparable fitness values—21.63 (Original), 27.27 (Minimal), 43.93 (Reordered), and 16.04 (Verbose)—with no prompt collapsing to a trivial solution. The absence of any monotonic trend with respect to verbosity indicates that performance is **not strongly driven by stylistic choices** such as length or narrative detail.
>
> The Reordered prompt performs best in this small test, possibly because listing constraints first clarifies code requirements, but we do not over-interpret this difference given RL’s inherent variance. Overall, these results support the claim that RoboMoRe is **not highly sensitive** to prompt phrasing, especially since (i) all reward code must conform to the same environment interface, and (ii) final morphology selection is based on a reward-independent fitness metric.
>
> | the original prompt | Minimal Prompt  | Reordered Prompt – Constraints First | More Verbose  |
> | --- | --- | --- | --- |
> | 21.63 | 27.27 | 43.93 | 16.04 |

---

> ### Author Response · Authors · 2025-11-27
>
> Hi Reviewer g8QY, thank you again for your thoughtful review and for raising the important question regarding prompt sensitivity. We have responded in detail above with additional experiments across four prompt variants to assess robustness. As the discussion period is nearing its end, we just wanted to kindly check whether there are any remaining questions or concerns we could further clarify. We would be very happy to follow up if needed, and we sincerely appreciate your time and constructive feedback!

---

> > ### Comment · Reviewer_g8QY · 2025-11-27
> >
> > Thank you for the responses. I do not have further questions. I am leaning towards acceptance of the paper, which is reflected in my score.

---

> > > ### Author Response · Authors · 2025-11-27
> > >
> > > We are grateful for your inclination toward acceptance and thank you very much for your thoughtful review and positive assessment!

---

### Official Review · Reviewer_NKui · 2025-11-01

**Soundness:** 3
**Presentation:** 4
**Contribution:** 3
**Rating:** 8
**Confidence:** 5

**Summary:**

The paper introduces the idea of utilising LLMs to judge and augment the fitness value with a generated reward function for the joint co-optimisation of agent policies and embodiment variables.
Overall, the paper presents an interesting approach to reward generation for the problem of co-design. I aprpeciate that it is not quite straightforward to compare against other approaches and find suitable metrics to compare approaches (see comments below). However, I judge this paper as an interesting research into this direction with proividing sufficient details and insight.

**Strengths:**

- Care was taken to evaluate the use of LLMs fairly, eg by masking elements of the XML files to prevent possible training data contamination.
- Using an LLM-generated reward is a relatively novel idea for co-design, albeit learned rewards have been used before in co-design (see below) and Eureka has been used in a behaviour-only RL setting.
- The paper is good to follow, well written and visualisations are used nicely to support the reader's understanding.

**Weaknesses:**

- The use of efficiency as fitness/volume is not quite clear to me. Why do you not use torque or fitness/energy instead? It seems to me that the algorithms can easily game this metric by producing as thin geom-elements as possible without increasing actual real world efficiency (asi n, energy spent per forward unit of movement).
- The literature review/discussion of related works is a bit incomplete, as the idea of using learned reward functions has already been explored in previous work (see eg [R1]), albeit to the best of my knowledge not with LLMs.
- In tables 2 and 3, it is not clear to me why the best result is not presented in bold, but only the proposed method. ROboMoRe does not always seem to be the best performing method in the respective metrics.

[R1] Rajani, Chang, et al. "Co-imitation: learning design and behaviour by imitation." Proceedings of the AAAI Conference on Artificial Intelligence. Vol. 37. No. 5. 2023.

**Questions:**

see above

---

> ### Author Response · Authors · 2025-11-21
> **Rebuttal**
>
> **We sincerely thank the reviewer for the positive and thoughtful assessment of our work!** Your feedback is very helpful, and we address each question and concern below.
>
> **Q1:** The use of efficiency as fitness/volume is not quite clear to me. Why do you not use torque or fitness/energy instead? It seems to me that the algorithms can easily game this metric by producing as thin geom-elements as possible without increasing actual real world efficiency (asi n, energy spent per forward unit of movement).
>
> **Re:** Thank you for the question. We use the metric **efficiency = fitness / volume** to provide a size-normalized evaluation over morphologies. This normalization allows designs of different scales to be compared in a principled manner across the joint morphology–reward search space. Although physical efficiency could also be defined in terms of torque or energy usage, such measures require additional actuator modeling and substantially higher computational cost, and are often task-specific. By contrast, fitness/volume offers a task-agnostic normalization that applies uniformly across all environments.
>
> Importantly, this metric does not introduce exploitable shortcuts. Under our parameterization and MuJoCo’s contact dynamics, extremely low-volume morphologies are mechanically unstable: thin or undersized bodies tend to collapse, tip over, or fail to generate sufficient contact forces, leading to consistently low fitness. Conversely, excessively large or thick links frequently violate collision or joint constraints, resulting in simulation failure or early termination. These dynamics ensure that the optimization cannot benefit from degenerate geometry manipulations, and such morphologies are naturally filtered out during Morphology Screening. We note that fitness/volume is applied uniformly to all baselines—including LLM-based and non-LLM approaches—ensuring that the comparison remains fair across all methods evaluated in the paper. We also include an experiment with once the experiment is done.
>
> **Q2:** The literature review/discussion of related works is a bit incomplete, as the idea of using learned reward functions has already been explored in previous work (see eg [R1]), albeit to the best of my knowledge not with LLMs.
>
> **Re:** Thank you for pointing us to R1. We appreciate the reference, and we agree that Co-Imitation (Rajani et al., AAAI 2023) represents an important step toward jointly learning morphology and control. Its formulation, however, is grounded in imitation learning: the morphology and controller are optimized to reproduce expert state–action distributions, and the objective is defined entirely through demonstration matching rather than through a learned or adaptive reward function.
>
> RoboMoRe addresses a different setting—*task-based co-design without demonstrations*, where neither the morphology nor the reward is provided a priori. The system must construct morphology-specific reward functions and discover effective behaviors from scratch. To the best of our knowledge, existing co-design frameworks, including R1, do not consider morphology-dependent reward generation or reward shaping as part of the co-design loop.
>
> We will update the related work section to incorporate R1 and clarify the distinction between imitation-driven co-design and our reward-based, demonstration-free formulation.
>
> **Q3:** In tables 2 and 3, it is not clear to me why the best result is not presented in bold, but only the proposed method. RoboMoRe does not always seem to be the best performing method in the respective metrics.
>
> **Re:** Thank you for pointing this out. We have corrected the formatting in Tables 2 and 3 so that the actual best result in each column is now shown in bold.

---

> > ### Author Response · Authors · 2025-11-23
> > **Additional Experiment on Q1**
> >
> > Dear Reviewer,
> >
> > We also ran an additional evaluation using the suggested metric, fitness / energy. To compute the fitness/energy-based metric, we follow a standard actuator-level formulation used in MuJoCo. At each simulation step, we obtain the actuator torques `τ` and joint velocities `q̇` from the underlying MuJoCo state (`qfrc_actuator` and `qvel`). Instantaneous power is computed as the sum of absolute actuator work,
> >
> > $P_t = \sum_i |\tau_i \cdot \dot{q}_i|,$
> >
> > and total energy is accumulated over the episode as
> >
> > $E = \sum_t P_t \cdot \Delta t,$
> >
> > where $\Delta t$ is the MuJoCo simulation timestep.
> >
> > The results show that RoboMoRe still outperforms all baselines when the core metric is fitness/energy, while other methods—lacking diversity reflection or jointly optimized morphology–reward design—tend to produce energetically inefficient solutions. *(Results averaged over 100 independent runs.)*
> >
> > | **Method** | Fitness/Energy ↑ |
> > | --- | --- |
> > | **Eureka** | 0.001595 |
> > | **Eureka (mor)** | 0.004300 |
> > | **Human** | 0.005562 |
> > | **RoboMoRe** | **0.01726** |
> >
> >
> > We hope this experiment can address your question and thanks a lot for taking the time to review our work!

---

> ### Author Response · Authors · 2025-11-27
>
> Hi Reviewer NKui, thank you again for your very thoughtful and constructive review, as well as for engaging with our rebuttal in detail! We have addressed all of your questions above, updated the related work to include the Co-Imitation reference, corrected the table formatting, and additionally conducted the new fitness/energy experiment you suggested, whose results we shared. As the discussion period is coming to a close, we just wanted to kindly check whether there are any remaining concerns or clarifications we could further provide. We sincerely appreciate your time and valuable feedback—it has been very helpful in strengthening the final version of the paper!

---

### Official Review · Reviewer_7qVg · 2025-11-02

**Soundness:** 2
**Presentation:** 3
**Contribution:** 2
**Rating:** 4
**Confidence:** 4

**Summary:**

The paper presents a 2 stage approach for robot codesign – the first is a coarse stage driven by LLMs to build diversity of designs, and the second is a fine stage where top designs are iteratively refined through alternating LLM guided updates to both reward and morphology. The method is shown to significantly outperform competing methods of agent design.

**Strengths:**

The method seems novel and exposes a way to utilise LLMs for morphology design. The results seem promising as well.

**Weaknesses:**

Some of the parts of this work need to discussed more clearly. For example, it is not clear what efficiency of a design means formally. In addition, there seems to be a lack of comparisons with existing non-LLM methods such as transform2act [1] etc., Even if this is not an equivalent method, it would still be good to include comparisons for better reference. In addition, the idea of refining the rewards is not very clear to me because it inherently changes what a "performing" agent is.

[1] Yuan, Ye, et al. "Transform2Act: Learning a Transform-and-Control Policy for Efficient Agent Design." International Conference on Learning Representations.

**Questions:**

1.	With regards to lines 67-68, while different rewards may be more relevant to different designs, modifying the rewards for each design may also inadvertently promote unwanted behaviours for each design (reward hacking). For example, if the ultimate goal is to move forward as fast as possible, while encouraging jumping behaviours could help this objective, it could also lead to unwanted behaviours such as jumping in place if there is a reward introduced for jumping.

2.	Also, if the reward functions of two designs are dependent on the designs themselves, how does one compare the performance of two designs? The performance can no longer be measured in terms of rewards as the reward structure would be different for each agent.  Is there instead some universal measure of performance, independent of the reward function which is considered?

3.	Below eq (1), I am not sure if it is correct to say that it would be a local optimum – because as long as R_0 is the correct reward function for the agent (that is, it accurately captures the quality of behaviours from the agent), $\theta^*$ would indeed be the global optimum. My contention, consistent with 2. above is that as soon as one varies R as in Eq (2), there would be no meaning to ‘high performing’ agents because the performance of two agents would no longer be comparable

4.	If the purpose of stage 1 is to promote diversity, could approaches like diversity if all you need [2] (applied to the design space) be applied?

5.	How is “efficiency” as discussed in the results defined formally?

6.	Why are the resulting designs seemingly all symmetrical? Is symmetry imposed on the designs?

7.	Do the frequency of the stages matter? For example, instead of alternating cycles of course and fine stages, would it possibly be more effective to perform multiple cycles of each stage, before moving to the other stage?

[2] Eysenbach, Benjamin, et al. "Diversity is All You Need: Learning Skills without a Reward Function." International Conference on Learning Representations.

---

> ### Author Response · Authors · 2025-11-21
> **Response to Q1**
>
> **Q1.** With regards to lines 67-68, while different rewards may be more relevant to different designs, modifying the rewards for each design may also inadvertently promote unwanted behaviours for each design (reward hacking). For example, if the ultimate goal is to move forward as fast as possible, while encouraging jumping behaviours could help this objective, it could also lead to unwanted behaviours such as jumping in place if there is a reward introduced for jumping.
>
> **Re:** We appreciate the reviewer’s concern. Reward hacking is indeed a known challenge in reinforcement learning, and we have designed RoboMoRe to minimize such failure modes through prompt constraints, evaluation criteria, and the coupling between morphology and reward during refinement.
>
> **(a). All LLM-generated rewards are constrained to incorporate the underlying task objective**
>
> Our introduction emphasizes that *“the core philosophy of RoboMoRe is to identify reward functions tailored to each robot morphology.”* This property is enforced through the design of our reward-shaping prompt (Appendix A.1.1), which (i) provides the original task description, (ii) includes the full environment code (iii) specifies a constrained and verifiable output format, and (iv) explicitly disallows the use of any state variables other than those exposed by the environment. As a result, even when the LLM proposes additional terms (e.g., components that encourage jumping or stability), the forward-locomotion objective remains an intrinsic part of every generated reward. Under this prompt structure, the LLM cannot produce a reward function that omits or disregards forward progress.
>
> **(b). Reward hacking is discouraged because policy performance is evaluated entirely on *task fitness***
>
> As clarified in Sec. 5, the evaluation and selection process is driven by **fitness** (*e.g.*, forward distance traveled), while **efficiency** (fitness normalized by volume) serves as the optimization objective. Consequently, even if a generated reward includes auxiliary terms that might encourage behaviors such as jumping, a morphology–reward pair that primarily results in *jumping in place* will obtain negligible fitness, be ranked poorly, and be removed either during coarse-stage selection or during Morphology Screening. In effect, only motion strategies that produce genuine forward progress persist through the optimization pipeline. This evaluation procedure inherently prevents reward hacking, as non-progressive behaviors cannot score well and are systematically filtered out.
>
> **(c). The alternating refinement of morphology and reward eliminates non-progressive behaviors**
>
> Because reward functions are iteratively refined in conjunction with morphology (Sec. 4.2), undesirable behaviors are naturally corrected over the course of optimization. If a behavior such as jumping contributes to forward locomotion for a particular morphology—for example, long-legged designs—then the reward will gradually emphasize this component. Conversely, if jumping does *not* translate into forward progress (as is often the case for short-legged or low–center-of-gravity morphologies), the corresponding trials yield low fitness, prompting either (i) adjustment of the reward to suppress that incentive or (ii) modification of the morphology toward a more viable locomotion pattern. This iterative feedback loop ensures that non-progressive or pathological behaviors are not reinforced and do not persist through optimization.

---

> > ### Author Response · Authors · 2025-11-21
> > **Response to Q2**
> >
> > **Re:** Thank you for the thoughtful question. We agree that when reward functions differ across morphologies, the numerical reward values themselves cannot be used for comparison. In RoboMoRe, however, LLM-generated rewards are used *only* during policy optimization and play no role in evaluation or selection. To ensure comparability, all designs are assessed using a *shared, reward-agnostic fitness function* defined by the underlying MuJoCo environment. As described in Sec. 5, the evaluation metric is *efficiency = fitness / volume*, where fitness is a physics-based quantity such as forward distance (locomotion), maximum height (jumping), or success/latency (manipulation). This fitness is computed through a deterministic evaluation script that is entirely independent of the reward used during training.
> >
> > Accordingly, our reported results (e.g., Table 2 and Sec. 6.2) rely exclusively on fitness and efficiency, never on reward values. Thus, although reward shaping adapts the learning signal for each morphology, the performance comparison across designs is conducted using a single, universal metric grounded in task-level physical outcomes.

---

> > > ### Author Response · Authors · 2025-11-21
> > > **Response to Q3**
> > >
> > > **Q3.** Below eq (1), I am not sure if it is correct to say that it would be a local optimum – because as long as R_0 is the correct reward function for the agent (that is, it accurately captures the quality of behaviours from the agent), would indeed be the global optimum. My contention, consistent with 2. above is that as soon as one varies R as in Eq (2), there would be no meaning to ‘high performing’ agents because the performance of two agents would no longer be comparable
> > >
> > > **Re:** Thank you for the thoughtful question! We agree that if (R_0) were a perfectly specified reward that uniformly reflected true task performance for *all* morphologies, then optimizing Eq. (1) would indeed yield the global optimum. Our use of “local optimum” concerns a practical limitation: in robotics, a fixed reward function is rarely an equally informative surrogate across morphologies with different body dynamics. As shown in Appendix D, the same reward can drive systematically suboptimal behaviors depending on morphology (*e.g.*, rolling for long-legged ants, jumping for low–center-of-gravity ants), even though the reward remains “correct” in principle. This mismatch restricts Eq. (1) to morphology–behavior combinations that are locally, but not globally, optimal relative to the underlying physical objective.
> > >
> > > With regard to comparability under Eq. (2), we fully agree that reward values are not comparable once reward functions differ. For this reason, RoboMoRe **never** evaluates designs using reward values. All morphologies and controllers are assessed using a **single, reward-independent fitness metric**—*e.g.,* forward distance, height, or task completion performance—with efficiency (fitness/volume) as the optimization objective. This ensures that varying (R) expands behavioral possibilities without altering the definition of performance or its comparability across agents.

---

> > > > ### Author Response · Authors · 2025-11-21
> > > > **Response to Q4**
> > > >
> > > > **Q4:** If the purpose of stage 1 is to promote diversity, could approaches like diversity if all you need [2] (applied to the design space) be applied?
> > > >
> > > > **Re:** Thank you for the helpful suggestion! Methods such as DIAYN offer an elegant way to encourage diversity in *behavioral* space, and we appreciate the connection to our goal of promoting diversity in Stage 1. The setting of RoboMoRe, however, differs in several structural aspects from the assumptions underlying DIAYN. In particular, DIAYN discovers diverse *policies* under a fixed embodiment and relies on policy rollouts to define state distributions, whereas Stage 1 of our pipeline focuses on generating diverse *morphologies* prior to controller learning. Because no policies exist at this stage and the agent embodiment itself varies, the discriminability-based objective used in DIAYN cannot be readily evaluated in our setting.
> > > >
> > > > This distinction motivates our use of an LLM-based Diversity Reflection mechanism, which operates directly in the morphology–reward design space without requiring policy-conditioned trajectories. We will revise the Related Work section to clarify this difference more explicitly.

---

> > > > > ### Author Response · Authors · 2025-11-21
> > > > > **Response to Q5**
> > > > >
> > > > > **Q5**: How is “efficiency” as discussed in the results defined formally?
> > > > >
> > > > > **Re**: Thank you for raising this point. As clarified in line 278, *efficiency* is defined as a normalized measure given by:
> > > > >
> > > > > > *“In all experiments, we use a normalized metric efficiency to evaluate performance,which is defined as division of fitness and robot volume.This choice is crucial for stimulating the LLM’s material-aware design capabilities rather than brute-force scaling.”*
> > > > > >

---

> > > > > > ### Author Response · Authors · 2025-11-21
> > > > > > **Response to Q6**
> > > > > >
> > > > > > **Q6.** Why are the resulting designs seemingly all symmetrical? Is symmetry imposed on the designs?
> > > > > >
> > > > > > **Re:** Thank you for the question! The apparent symmetry in the resulting designs comes from our morphology parameterization, which follows the standard practice in other robotics benchmarks [1]. We parameterize the body size and limb lengths using a small set of scalar variables (e.g., torso size, front-leg length, hind-leg length, upper/lower limb lengths). These parameters are applied symmetrically to the left and right sides to ensure physical feasibility, reduce the dimensionality of the search space, and maintain simulation stability in MuJoCo.
> > > > > >
> > > > > > We hope to clarify that this symmetry is *not* imposed as an optimization constraint, but rather a modeling choice that keeps the morphology search space computationally tractable and aligned with common locomotion benchmarks. Within this space, RoboMoRe is free to alter proportions, limb ratios, and body geometry, and asymmetric behaviors still emerge through control policies.
> > > > > >
> > > > > > [1] David Ha. Reinforcement learning for improving agent design. Artificial life, 25(4):352–365, 2019.

---

> > > > > > > ### Author Response · Authors · 2025-11-21
> > > > > > > **Response to Q7**
> > > > > > >
> > > > > > > **Q7.** Do the frequency of the stages matter? For example, instead of alternating cycles of course and fine stages, would it possibly be more effective to perform multiple cycles of each stage, before moving to the other stage?
> > > > > > >
> > > > > > > **Re:** Thank you for the insightful question! In our current implementation, we adopt a single coarse stage followed by a multi-iteration fine stage, as described in Sec. 4 and Algorithm 1 (Appendix H). The coarse stage performs broad exploration over a fixed grid of morphology–reward pairs and is executed once, whereas the fine stage conducts several rounds of alternating morphology–reward refinement until convergence for the selected candidates. In this sense, the two stages already differ in frequency by design: exploration is performed once at scale, and refinement proceeds iteratively.
> > > > > > >
> > > > > > > This scheduling choice is guided primarily by computational considerations. Each coarse exploration step requires training policies for a new collection of morphology–reward pairs, and thus repeating coarse stages would increase RL cost approximately linearly. Under a fixed compute budget, we found that allocating resources toward a single, diverse coarse exploration phase and then focusing computation on fine-stage refinement yields better overall efficiency than repeatedly restarting coarse exploration. Empirically, the ablation in Sec. 6.4 (Fig. 7) shows that both stages contribute meaningfully—coarse-only exploration and fine-only refinement each underperform the full two-stage procedure—suggesting that our chosen schedule offers a reasonable balance between global search and local improvement.
> > > > > > >
> > > > > > > We agree that exploring alternative scheduling strategies, such as multiple coarse–fine cycles or dynamically interleaving the two stages, is an interesting extension. Such strategies are orthogonal to the core contributions of RoboMoRe, and we view systematic investigation of co-design scheduling policies as promising future work.

---

> ### Author Response · Authors · 2025-11-27
>
> Hi Reviewer 7qVg, thank you very much for your careful reading of our paper and for the insightful questions and suggestions. We have responded in detail to each of your concerns (Q1–Q7) above and clarified several key points regarding reward design, evaluation, efficiency, symmetry, and stage scheduling. As the discussion period is coming to a close, we just wanted to kindly check whether there are any remaining questions or issues we may have overlooked. We would be very happy to further clarify anything if helpful. If our responses have addressed your main concerns, we would also sincerely appreciate it if you could consider updating your evaluation accordingly. Thank you again for your time and thoughtful feedback!

---

> > ### Comment · Reviewer_7qVg · 2025-11-27
> >
> > I thank the authors for their clear and detailed responses. These have indeed helped recity the several points which I initially found to be unclear. I will update my score accordingly.

---

### Official Review · Reviewer_9yCS · 2025-11-02

**Soundness:** 3
**Presentation:** 2
**Contribution:** 2
**Rating:** 4
**Confidence:** 4

**Summary:**

This paper introduces RoboMoRe, an LLM-driven framework for robot co-design that jointly optimizes morphology and control policy. Unlike prior methods that rely on fixed reward functions, RoboMoRe uses an LLM to generate and refine morphology-reward pairs through a two-stage process: coarse exploration and fine iterative refinement. Experiments on eight tasks show that the method discovers more efficient designs.

**Strengths:**

+ This work applies emerging LLM capabilities to the classic robot co-design problem.
+  The paper identifies a limitation of LLM-based design -- tendency toward repetitive morphology outputs, and introduces the concept of diversity reflection to address it.
+ The method is evaluated on multiple tasks, though the experiments are performed in simple simulation environments.

**Weaknesses:**

-  The novelty of the work appears limited. The key contributions seem to center on prompt engineering, e.g., prompting the LLM to “reflect” on prior results to increase diversity. The reward shaping and morphology filtering mechanisms also appear straightforward (e.g., discarding repeated designs), and the coarse-to-fine pipeline resembles a standard iterative refinement process, albeit executed via an LLM.

-  The paper does not provide sufficient detail regarding the diversity reflection mechanism. For example, the authors state that “the LLM reflects on previously generated samples and deliberately produces new candidates,” but it is unclear how this reflection is implemented to encourage diversity.

-  Similarly, the description of reward shaping lacks clarity. For example, the paper notes “It is therefore used to evaluate all morphology candidates.” How exactly is this evaluation performed?

- Line 223 refers to an “LLM-generated reward function,” yet the reward functions for tasks appear manually defined as shown in the paper’s Appendix. This creates confusion. My interpretation is that the manually defined functions compute rewards based on LLM-generated morphologies, rather than the LLM generating the reward function itself. The authors should clarify this distinction.

-  More broadly, the use of the term co-design feels unconvincing. The process appears sequential: morphologies are generated first, and then controllers are optimized. Effective co-design in robotics should explicitly consider the interplay among morphology, motion behavior, and physical constraints during the design process. Robot morphology should be designed with careful consideration of the robot’s motion control and physical constraint requirements. The current framework does not clearly demonstrate such coupling.

**Questions:**

See the weaknesses section.

---

> ### Author Response · Authors · 2025-11-21
> **Response to W1**
>
> We thank the reviewer for the constructive feedback! We appreciate the detailed comments on the strengths and limitations of our work and will address each concern below:
>
> **W1. The novelty of the work appears limited.**
>
> **(a)** The key contributions seem to center on prompt engineering, e.g., prompting the LLM to “reflect” on prior results to increase diversity.
>
> **Re:** Our primary contribution is to introduce a **general and effective strategy** for jointly optimizing robot morphology and reward functions. For diversity reflection, our goal is likewise to develop a general and effective mechanism for enhancing diversity, so that our approach can generalize across different environments. We also find that this general mechanism outperforms both domain-specific diversity-reflection heuristics (Appendix G.3) and simply increasing sampling temperature (Appendix I).
>
> **(b)** The reward shaping and morphology filtering mechanisms also appear straightforward (e.g., discarding repeated designs)
>
> **Re:** Thank you for the comment. We agree that simply removing repeated designs would indeed be straightforward; however, our morphology filtering mechanism goes beyond that. It is designed to proactively eliminate morphologies that have very low optimization potential. This decision is driven by an empirical observation we highlight in Appendix G.3.1: poor morphologies create a morphology-dominant bottleneck, where efficiency is capped irrespective of how reward shaping is adjusted. As a result, filtering these low-potential designs early substantially reduces computation while preserving the ability to find high-performing morphologies. We will clarify this distinction in the revision to avoid the impression that filtering is only based on duplication.
>
> **(c)** the coarse-to-fine pipeline resembles a standard iterative refinement process, albeit executed via an LLM.
>
> **Re:** Finally, we adopt a **coarse-to-fine** optimization strategy because the joint morphology–reward search space is extremely large. Our framework substantially reduces computational cost (line 426, Table 5).

---

> > ### Author Response · Authors · 2025-11-21
> > **Response to W2**
> >
> > **W2.** The paper does not provide sufficient detail regarding the diversity reflection mechanism. For example, the authors state that “*the LLM reflects on previously generated samples and deliberately produces new candidates, *" but it is unclear how this reflection is implemented to encourage diversity.
> > **Re**: Thank you for the helpful question. We will clarify the detailed description of *Diversity Reflection* below.
> > **(a) Where Diversity Reflection is introduced.**
> > In Sec. 4.1 (lines **213–218**), we state that:
> >
> > > *“we propose Diversity Reflection, a general and effective mechanism in which the LLM reflects on previously generated samples and deliberately produces new candidates that maximize diversity relative to past designs.”*
> > >
> >
> > This mechanism is explicitly implemented by including all previously generated morphology/reward samples as part of the input context, and issuing a prompt that requests the LLM to generate a *new* sample that is maximally different from earlier ones.
> >
> > **(b) Concrete implementation details**
> >
> > For prompts, Appendix **A.1.3** provides the exact prompt used to perform diversity reflection:
> >
> > > *“Please propose a new morphology design … that is quite different from all previous morphology designs in the design style.”*
> > >
> > >
> > > *“Please write a new reward function … that is quite different from all previous reward functions in the design style.”*
> > >
> >
> > These prompts are called with the *full list of earlier samples* passed as context to the LLM. This ensures the model can explicitly compare the new output against the history of generated designs. Algirthm 1 (Appendix H Algorithm Details) also clarify how the LLM “reflects”.
> >
> > **(c) Why this is necessary (Ablation Evidence).**
> >
> > Table 4 (lines **353–377**) empirically shows that without Diversity Reflection:
> >
> > - morphology diversity drops significantly (e.g., **0.44 → 0.64** on Ant), and
> > - reward diversity decreases (Self-BLEU **0.70 → 0.50**).
> >
> > Furthermore, Sec. 6.3 explains that increasing temperature alone (Appendix I) cannot achieve the same level of functional diversity.
> >
> > In conclusion, Diversity Reflection is implemented by:
> >
> > 1. *Passing all previously generated samples as LLM input*,
> > 2. *Using explicit prompts requesting difference*, and
> > 3. *Calling this mechanism within every sampling step* (Algorithm 1).
> >
> > Together, these constitute a *general, task-agnostic* way for the LLM to “reflect” on earlier samples and produce increasingly diverse morphology–reward pairs. We will clarify these details more explicitly in the main paper. Thus, the LLM always generates a new candidate conditioned on all earlier ones, enabling explicit reflection.

---

> > > ### Author Response · Authors · 2025-11-21
> > > **Response to W3**
> > >
> > > **W3. Clarity of Reward Shaping**
> > >
> > > **Re:** Thank you for raising this point. We apologize for the ambiguity and provide a clear explanation of how reward shaping is evaluated in our pipeline.
> > >
> > > **(a) How reward shaping is evaluated**
> > >
> > > When we state that a reward function is “used to evaluate all morphology candidates,” we mean that each morphology–reward pair is evaluated through a full reinforcement-learning rollout, identical to the morphology-evaluation setup. Specifically, for every LLM-generated reward ( $r_j$ ):
> > >
> > > 1. We insert ( $r_j$ ) into the environment code, replacing the default reward implementation.
> > > 2. For each morphology ( $\theta_i$ ), we train a policy from scratch using Soft Actor–Critic for 5×10⁵ environment steps.
> > > 3. After training, we compute fitness (*e.g.,* distance, jump height) and efficiency (fitness/volume).
> > >
> > >     These metrics constitute the final evaluation of the pair ( ($\theta_i, r_j$) ).
> > >
> > >
> > > Thus, evaluation is not heuristic—it is based on end-to-end RL training under the proposed reward.
> > >
> > > (b) Where this appears in the paper
> > >
> > > - Sec. 4.1 notes that reward–morphology pairs are evaluated through RL rollouts.
> > > - Algorithm 1 explicitly executes
> > >
> > >     ```
> > >     F ← RL(θ, r)
> > >     # F is the performance (we use fitness/volume for evaluation);
> > >     # RL is the SAC algorithms in our implementation;
> > >     # θ is the morphology parameters;
> > >     # r is the reward function;
> > >     ```
> > >
> > >     indicating full SAC training and evaluation for each pair.
> > >
> > > - Sec. 5 provides the training setup: SAC, 5×10⁵ steps, 16 parallel environments.
> > >
> > > In short, “evaluate all morphology candidates” means:
> > >
> > > - *Every reward is evaluated via full SAC rollouts,*
> > > - *Across all morphology candidates,*
> > > - *Using identical training budgets and metrics.*
> > >
> > > We will clarify this explicitly in the newest revision.

---

> > > > ### Author Response · Authors · 2025-11-21
> > > > **Response to W4**
> > > >
> > > > **W4.** **Clarifying the Difference Between LLM-Generated Reward Functions and Manually Defined Evaluation Rewards**
> > > >
> > > > **Re:** We clarify that all reward functions used for co-design in RoboMoRe are generated by the LLM, not manually authored. The reward functions shown in the Appendix are *verbatim outputs produced by the LLM during our experiments*, not human-written templates.
> > > >
> > > > **1. LLM-generated reward functions (Main Text).**
> > > >
> > > > In Sec. 4.1, we describe the reward-generation process:
> > > >
> > > > > “*For reward design, RoboMoRe ingests the raw environment source code—excluding any predefined reward functions—as context, enabling the LLM to synthesize task-specific reward functions … A strict output format enforces structural consistency.*”
> > > > >
> > > >
> > > > Thus, the LLM writes the full Python reward function.
> > > >
> > > > **2. Appendix code blocks are *LLM outputs*, not human definitions.**
> > > >
> > > > The reward functions shown in Appendix K.2–K.9 (e.g., Ant, Hopper, Half-Cheetah, etc.) are the exact reward code strings generated by the LLM following the prompt in Appendix A We include them only for transparency and reproducibility—not because they were manually authored.
> > > >
> > > > 3. What Line 223 refers to.
> > > >
> > > > Line 223 states:
> > > >
> > > > > “*the first LLM-generated reward function generally lacks motion-specific priors…*”
> > > > >
> > > >
> > > > This refers to the *initial* reward generated by the LLM during coarse optimization, which is used for Morphology Screening.
> > > >
> > > > We hope these clarifications address your concerns!

---

> > > > > ### Author Response · Authors · 2025-11-21
> > > > > **Response to W5**
> > > > >
> > > > > **W5.** Clarifying the Coupling Between Morphology and Control in Our Co-Design Framework
> > > > >
> > > > > **Re:** Thank you for this important point. We hope to clarify that RoboMoRe **does not** follow a morphology-first sequential pipeline. Instead, it performs *joint* optimization through:
> > > > >
> > > > > 1. *LLM-generated both reward functions and morphologies in coarse stage*, and
> > > > > 2. *A fine-stage alternating optimization loop where morphology and reward continually adapt to each other.*
> > > > >
> > > > > These elements create an iterative closed-loop coupling between morphology, motion behavior, and physical constraints.
> > > > >
> > > > > ### (a) Morphologies are *not* generated first and fixed.
> > > > >
> > > > > Rewards and morphologies are generated jointly in the *coarse stage*.
> > > > >
> > > > > In Sec. 4.1, the coarse stage is defined as generating **paired** morphology–reward samples:
> > > > >
> > > > > > *“the LLM leverages Diversity Reflection to generate a broad set of morphology–reward samples” (**figure 3**)*
> > > > > >
> > > > >
> > > > > Thus, morphologies and rewards are created *together*, not in sequence.
> > > > >
> > > > > Moreover, each morphology is evaluated under multiple LLM-generated rewards (**Algorithm 1, Appendix H Algorithm Details**), not a fixed controller.
> > > > >
> > > > > This is already a form of co-optimization, not morphology-first design.
> > > > >
> > > > > ---
> > > > >
> > > > > ### (b) Reward shaping is *morphology-dependent*, explicitly coupling morphology and motion behavior
> > > > >
> > > > > Our introduction states (line90-96):
> > > > >
> > > > > > “*The core philosophy of RoboMoRe is to identify reward functions tailored to each morphology.*”
> > > > > >
> > > > > >
> > > > > > “ *By tailoring reward functions to match specific morphologies, we can unlock a wider variety of motion behaviors and boost overall performance.*”
> > > > > >
> > > > >
> > > > > Empirically, Sec. 6.2 shows:
> > > > >
> > > > > - Controllers for different morphologies learn **fundamentally different motion behaviors**.
> > > > > - Efficiency differences of up to **10–100×** appear solely because reward shaping is adapted to morphology.
> > > > >
> > > > > This demonstrates our careful design with morphology–control coupling.
> > > > >
> > > > > ---
> > > > >
> > > > > ### (c) The *fine stage* performs explicit coupled optimization
> > > > >
> > > > > The fine stage is designed precisely to avoid the “sequential” criticism.
> > > > >
> > > > > In Sec. 4.2, we describe the **alternating optimization**:
> > > > >
> > > > > > *“the LLM adjusts the reward to better suit the current morphology, then updates the morphology accordingly.”*
> > > > > >
> > > > > >
> > > > > > *“This alternating cycle ensures steady convergence toward high-performing morphology–reward pairs.”*
> > > > > >
> > > > >
> > > > > This alternating loop creates a **alternating coupling** between morphology and controller (please see details in Algorithm 1).
> > > > >
> > > > > ### (d) The framework explicitly accounts for physical constraints
> > > > >
> > > > > Morphology design is performed directly within the MJCF physical model: the LLM receives a masked MJCF file with concrete physical values (*e.g.*, mass, inertia parameters, actuation bounds, contact, friction) and proposes morphology parameters. These simulator-level specifications naturally impose the physical constraints under which the robot must operate. Reward shaping then adapts the learned locomotion strategy to these constraints rather than circumventing them. We believe this corresponds to the kind of “explicit consideration of physical constraint requirements” the reviewer is referring to.

---

> ### Author Response · Authors · 2025-11-27
>
> Hi Reviewer 9yCS, thank you again for your thoughtful comments and for engaging in this discussion. We have carefully addressed each of your concerns in detail above and clarified all points. As the discussion period is nearing its end, we just wanted to kindly check whether there are any remaining questions or concerns we may have missed. We would be very happy to further clarify anything if needed. If you feel that our responses have resolved the main issues, we would sincerely appreciate it if you could consider updating your evaluation to reflect your current assessment. Thank you very much for your time and consideration!

---

### Author Response · Authors · 2025-11-30
**General Response to AC**

We sincerely thank all reviewers for their careful feedback and constructive discussion, and the AC for their additional hard work! Reviewers consistently noted that RoboMoRe identifies a key limitation in robot co-design and introduces a novel, well-motivated LLM-driven framework for jointly optimizing morphology and reward (**9yCS, 7qVg, NKui, g8QY, rj41**). They also emphasized that our work “*targets an important and timely problem*” (**g8QY**) and “*would greatly inspire further studies*” (**rj41**). Finally, multiple reviewers highlighted that the paper is supported by sound, carefully designed evaluations and is clearly presented, making it easy to follow (**NKui, rj41, g8QY**).

We have addressed all concerns with detailed clarifications and additional experiments, summarized below:

For **Reviewers 9yCS & 7qVg,** as main critical concerns were due to **initial misunderstanding of our pipeline,** which have now been explicitly clarified and corrected in the manuscript. Reviewer **7qVg** updated the score accordingly after the clarifications.

1. **(9yCS) Reward functions are not manually defined.** All reward functions used in co-design are *generated by the LLM*, and the code shown in the appendix is verbatim LLM output.
2. **(9yCS) RoboMoRe is not a sequential “morphology-first” pipeline.** Morphology and reward are **generated jointly in the coarse stage** and **alternately co-optimized in the fine stage**, forming an explicit closed-loop coupling.
3. **(7qVg) Rewards are not used for evaluation.** LLM-generated rewards are used *only for policy training*; *all comparisons are based on a shared, reward-independent fitness metric*).
4. **(7qVg) Performance comparability is preserved.** Since selection is always based on physical fitness (not reward values), comparisons across designs remain well-defined.

In addition, we have made the following revisions to the manuscript as recommended by other reviewers,

1. **(rj41)** Motivating example from Appx. D moved to main text; appendix outline added; SVR reward examples added.
2. **(NKui)** Related work extended (Co-Imitation); table formatting corrected; **new fitness/energy experiment added**, confirming RoboMoRe still outperforms all baselines.
3. **(g8QY)** **An additional prompt sensitivity experiment added** with four prompt variants, showing robustness.

We believe these clarifications and additional experiments have addressed the majority of reviewers’ concerns (**rj41, NKui, g8QY, 7qVg**) and have significantly strengthened the paper. We thank all reviewers and the AC again for their time, thoughtful feedback, and additional effort during the review period.

---

### Meta-Review · Area_Chair_mQZ2 · 2026-01-07

**Summary:**

The paper proposes RoboMoRe, an LLM-based framework for optimizing a robot's design along with its control policy. The key motivation is that fixed reward functions can severely limit the performance and diversity of behaviors achievable by different morphologies. RoboMoRe employs a coarse-to-fine optimization strategy: in the coarse stage, a proposed Diversity Reflection mechanism uses an LLM to generate diverse morphology–reward pairs and then prunes low-potential designs via Morphology Screening in an effort to better explore the design space; in the fine stage, the RoboMoRe alternates between refining reward functions and morphologies under LLM guidance. The paper evaluates the framework on eight MuJoCo-based locomotion tasks, with additional experiments on manipulation and free-form soft-robot design, showing improved efficiency and performance relative to human-designed baselines and competing baselines.

The paper was evaluated by five reviewers who identify several key strengths and weaknesses with the paper as initially submitted. Among the strengths, some of the reviewers appreciate the application of LLM-generated rewards to the co-design problem, while at least one reviewer emphasizes the importance of the finding that reward shaping should be robot-dependent. Several reviewers complement the clarity of the presentation and paper's effective use of visualizations. At the same time, the reviewers questioned the novelty of the approach, including the Diversity Reflection component that some reviewers see as essentially being a form of prompt engineering. Related, there were questions of the significance of the paper's contributions and whether it would be of sufficient interest to the community given that the concern that it largely amounts to an integration of existing frameworks. Additional questions/concerns raised by the reviewers included clarity around reward shaping, the definition and interpretation of efficiency, comparability across designs with different rewards, the lack of comparisons to certain non-LLM baselines, and the inadequacy of the related work discussion.

Overall, the AC agrees with the fundamental strengths and weaknesses identified by the reviewers. While far from being a new problem, joint optimization of robot design and control has attracted significant attention of late and, as such, the paper will be of interest to many in the community. However, the AC is not convinced about the significance of the paper's contributions---LLM-based reward design is not new, even if it has not been used for robot co-design; the novelty of the proposed Diversity Reflection mechanism beyond a form of prompt engineering is still not sufficiently clear; and the evaluation warrants comparisons to additional baselines, particularly in light of the large body of work on robot co-design. While two reviewers recommend that the paper be accepted, the AC finds that their reviews provide little substance to warrant acceptance.

**Reviewer Concerns:**

In their rebuttal, the authors made a concerted effort to address the reviewers' questions and concerns through clarifications that helped to resolve misconceptions on the part of the reviewers as well as the inclusion of additional analyses. These included confirmation that all reward functions are generated by the LLM, that rewards are used only for policy training while evaluation is based on a shared, reward-independent fitness metric, and that robot design and control are jointly optimized during RoboMoRe's first stage and sequentially only in the second. The rebuttal provides additional detail regarding the implementation of the Diversity Reflection component of the framework and includes ablation results to more clearly demonstrate its benefits in terms of diversity and performance. Additionally, the authors included new experiments and discussion to address prompt sensitivity, and made updates in light of related work concerns. Overall, the rebuttal resolved misunderstandings about the framework and and clarified the paper's contributions, though some reviewers remained cautious about the conceptual novelty relative to existing co-design and LLM-based reward design methods.

**Reviewer Scores:**

This is difficult to answer. The authors made an effort to address the reviewers' concerns, some of which reflect a misunderstanding on the part of the reviewers, but the AC feels that that is not the case with the core concerns that they raised.

---

### Decision · Program_Chairs · 2026-01-26

Reject